# De4D-SLAM: Gradient-Isolated Static-Dynamic Decoupling for Monocular SLAM in Dynamic Environments

Zhicheng Fan [1]    Zitong Wu [1]    Zhaoxing Fan [1]    Xiao Zhang [1]    Biao Hou [1]    Bo Ren [1]

## Abstract

Conventional dynamic SLAM approaches typically treat dynamic objects as outliers based on pre-defined categories, creating perceptual blind spots that limit the comprehensive environmental perception required for embodied agents. Although integrating Gaussian Splatting into SLAM enables holistic scene representation, it introduces an optimization paradox: without categorical priors, flexible dynamic primitives rapidly overfit static residuals. This phenomenon undermines the self-supervised error signals necessary for distinguishing motion. In response, we present De4D-SLAM, a novel framework designed for decoupled 4D reconstruction from monocular video. Our approach features a Gradient-Isolated Decoupling strategy, which leverages static reconstruction residuals to supervise a Spatially-Aware Kolmogorov-Arnold Network (SA-KAN), ensuring robust, category-agnostic motion segmentation. Additionally, we propose a Flow-Induced Initialization prior to stabilize the non-convex optimization of 4D Gaussian primitives using dense optical flow. Extensive evaluations on the TUM and Bonn benchmarks demonstrate that De4D-SLAM achieves competitive performance in both tracking and dynamic reconstruction, successfully reconciling the tension between robust localization and high-fidelity 4D mapping.

## 1. Introduction

Simultaneous Localization and Mapping (SLAM) constitutes the perceptual backbone of embodied AI. For autonomous agents operating in human-populated environments, perceiving dynamics is a safety-critical necessity.

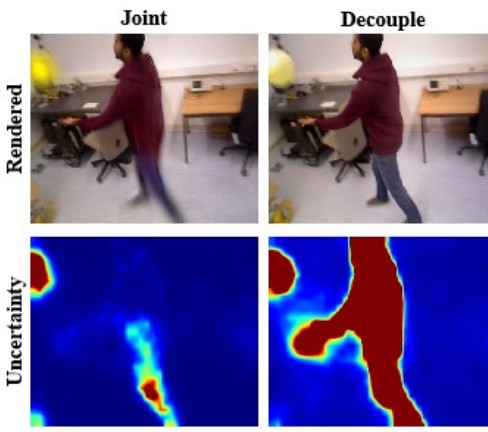

*Figure 1.* **The Optimization Paradox in Unsupervised Dynamic SLAM. Left (Joint):** Naive joint optimization leads to *mask collapse*, where flexible dynamic primitives overfit the static background (blue), causing motion blur. **Right (Decoupled): De4D-SLAM** uses Gradient-Isolated Decoupling to robustly identify motion (red) via **SA-KAN**, enabling sharp, holistic 4D reconstruction.

However, traditional SLAM systems, typically built upon sparse features (Mur-Artal et al., 2015) or dense photometric alignment (Teed & Deng, 2021), are predicated on a *static world assumption*. Consequently, dynamic entities violate geometric constraints, causing these systems to fail by treating moving objects as erroneous landmarks, leading to tracking drift and map corruption.

To mitigate these failures, contemporary approaches predominantly adopt an exclusion-based paradigm (Wang et al., 2024), treating dynamics as noise. While stabilizing localization, this creates dangerous perceptual blind spots. This deficiency is particularly acute in monocular setups, our primary focus, where masking dynamics without active depth sensors (e.g., LiDAR) creates a *perceptual void*, effectively erasing physical obstacles from the map.

We contend that robust perception requires shifting to holistic reconstruction: simultaneously maintaining a static map for localization and dynamic representations to fill the void. While recent works like 4DGS-SLAM (Li et al., 2025) and

[1]School of Artificial Intelligence, Xidian University, Xi'an, China. Correspondence to: Zitong Wu <wuzitong@xidian.edu.cn>, Biao Hou <avcodec@163.com>.

D4DGS-SLAM (Sun et al., 2025) explore 4D Gaussian Splatting, they often rely on external priors (e.g., YOLO) or RGB-D inputs. This reliance limits them to closed-set categories or specific hardware. Achieving this holistically in a monocular setting introduces a fundamental **Optimization Paradox**: unsupervised motion discovery relies on static reconstruction residuals. However, without explicit categorical supervision, flexible dynamic primitives rapidly overfit these residuals during naive joint optimization. This extinguishes the supervisory signal required for segmentation, causing the system to fail in distinguishing motion.

To resolve this paradox, we present **De4D-SLAM**, a dual-stream framework explicitly designed for Gradient-Isolated Static-Dynamic Decoupling. As demonstrated in Figure 1, this architecture prevents the mask collapse typical of joint optimization. Crucially, the learned uncertainty map exhibits a sharp contrast: it assigns high values to explicitly highlight dynamic entities, while rigorously suppressing static background regions toward zero. This distinct separation ensures that the dynamic stream is activated exclusively for valid motion, thereby successfully resolving the optimization paradox. Specifically, we leverage static residuals to supervise a Spatially-Aware Kolmogorov-Arnold Network (SA-KAN), which acts as a soft gate to activate the dynamic stream only upon static failure. This mechanism strictly isolates background gradients. Furthermore, we introduce a Flow-Induced Initialization to stabilize the non-convex optimization using dense optical flow. The project repository is available at https://github.com/zhichengf/De4D-SLAM.

Our main contributions are summarized as follows:

- We identify the **Optimization Paradox** in unsupervised dynamic SLAM and propose a **Gradient-Isolated Decoupling** strategy within a dual-stream architecture to robustly separate static background and dynamic entities without categorical priors.

- We introduce **SA-KAN**, a lightweight hybrid architecture that integrates depthwise convolutions with Kolmogorov-Arnold Networks to infer spatially coherent motion probability fields from visual features.

- We propose a **Flow-Induced Initialization** prior that utilizes dense optical flow to resolve the non-convexity of monocular 4D primitive optimization.

- We demonstrate that our system achieves competitive performance on the TUM and Bonn benchmarks in both camera tracking (ATE) and dynamic scene rendering (PSNR/LPIPS), validating the mutual benefit of integrated tracking and mapping.

**Conflict of Interest Disclosure.** The authors declare that this research was conducted in the absence of any commercial, financial, or personal relationships that could be construed as a potential conflict of interest. All authors are affiliated with academic institutions, and no funding or model evaluations in this work were subject to third-party commercial intervention.

## 2. Related Work

**Geometric and Semantic Dynamic SLAM.** Classical visual SLAM relies on static assumptions (Mur-Artal & Tardós, 2017; Engel et al., 2018), often drifting in dynamic environments due to geometric violations. Geometric methods mitigate this by filtering outliers via epipolar constraints (Scona et al., 2018) or scene flow (Yin et al., 2023), yet they discard dynamic content. Semantic approaches leverage deep learning: CubeSLAM (Yang & Scherer, 2019) tracks rigid bounding boxes but misses non-rigid deformations; dense methods like DynaSLAM (Bescos et al., 2018), Mask-SLAM (Kaneko et al., 2018), and DS-SLAM (Yu et al., 2018) mask dynamic regions or inpaint backgrounds. While VDO-SLAM (Zhang et al., 2020) tracks dynamic points, these methods remain limited by closed-set detectors. In contrast, De4D-SLAM achieves dense, class-agnostic reconstruction of complex non-rigid motions via gradient-isolated disentanglement. Unlike recent open-set 3D mapping approaches such as ConceptFusion (Jatavallabhula et al., 2023), which use foundation models primarily for semantic querying, we leverage robust DINOv2 features (Oquab et al., 2024) to enforce geometric consistency for category-agnostic dynamic separation without relying on categorical priors.

**Neural Implicit and Gaussian SLAM.** NeRF (Mildenhall et al., 2021) and 3D Gaussian Splatting (3DGS) (Kerbl et al., 2023) have revolutionized dense mapping (Sucar et al., 2021; Zhu et al., 2022; Wang et al., 2023; Sandström et al., 2023). Systems like MonoGS (Matsuki et al., 2024), Splat-SLAM (Keetha et al., 2024), and GS-SLAM (Yan et al., 2024) integrate 3DGS, though often assuming static scenes (Peng et al., 2024). Addressing dynamics, WildGS-SLAM (Zheng et al., 2025) filters features via learned uncertainty. However, while WildGS rejects dynamics as noise, we reconceptualize uncertainty as an active supervisory signal. Specifically, we use static residuals to supervise our SA-KAN, a soft gating mechanism within a dual-stream architecture that selectively recruits dynamic primitives for holistic reconstruction.

**4D Scene Representations.** Monocular 4D reconstruction is ill-posed, addressed by continuous deformation fields (Pumarola et al., 2021; Park et al., 2021; Fang et al., 2022) or discrete temporal parameters (Yang et al., 2024; Wu et al., 2024; Luiten et al., 2024; Lin et al., 2024). Notably, FreeTimeGS (Wang et al., 2025) offers efficient modeling via explicit velocity. However, adapting these to

SLAM is challenging. Current solutions often rely on external priors like YOLO (4DGS-SLAM (Li et al., 2025)) or RGB-D inputs (D4DGS-SLAM (Sun et al., 2025)). De4D-SLAM tackles monocular reconstruction without such constraints. We achieve robust disentanglement via Gradient-Isolated Decoupling and stabilize optimization using Flow-Induced Initialization to handle complex non-rigid motions.

## 3. Method

We present **De4D-SLAM**, a unified framework for dense monocular SLAM in dynamic environments. Unlike conventional approaches relying on pre-trained closed-set segmenters, our system aims to reconstruct arbitrary motion in a class-agnostic manner. As illustrated in Figure 2, the pipeline comprises a tracking frontend and a mapping backend. The frontend, built upon dense bundle adjustment (Teed & Deng, 2021), leverages dense optical flow and recurrent updates to robustly estimate camera poses $\mathbf{T} \in SE(3)$ and depth maps $\mathbf{D}$. The backend implements our proposed Gradient-Isolated Dual-Stream Architecture, which effectively decomposes the scene into a *Static Gaussian Stream* $\mathcal{G}_s$ for the rigid background and a *Dynamic 4D Gaussian Stream* $\mathcal{G}_d$ for moving entities. To orchestrate this separation, we introduce the Learnable Disentanglement Module (LDM). As shown in the system overview, this module encapsulates a frozen feature extractor and our core Spatially-Aware Kolmogorov-Arnold Network (SA-KAN), functioning as a soft gating mechanism to adaptively modulate gradient flows and rendering contributions based on geometric and perceptual consistency.

### 3.1. Decoupled Scene Representation

We formulate the scene as a composite set of Gaussian primitives $\mathcal{G} = \{g_i\}$, explicitly partitioned by a binary state $s_i \in \{\text{static}, \text{dynamic}\}$. This design facilitates holistic scene reconstruction within a single pipeline, while strictly enforcing distinct kinematic constraints to ensure decoupling.

**Static Primitives ($s_i = $ static).** Representing the rigid background, these primitives are parameterized by standard 3D Gaussian attributes: mean position $\mu_i$, covariance $\Sigma_i$, color $c_i$, and opacity $\alpha_i$. We enforce rigidity and persistence by constraining their velocity to zero and temporal extent to infinity: $\mathbf{v}_i = \mathbf{0}, \quad \sigma_{t,i} \to \infty$.

**Dynamic Primitives ($s_i = $ dynamic).** To capture object motion, these primitives are augmented with 4D dynamics. Adopting the efficient parameterization from FreeTimeGS (Wang et al., 2025), we model their position evolution $\mu_i(t)$ linearly based on a learnable velocity vector $\mathbf{v}_i$:

$$\mu_i(t) = \mu_{i,0} + \mathbf{v}_i \cdot (t - t_{i,0}), \tag{1}$$

where $\mu_{i,0}$ denotes the canonical position at timestamp $t_{i,0}$. Similarly, their visibility is modulated by a temporal opacity function $\alpha_i(t)$ to handle topological changes (e.g., objects entering or leaving the view):

$$\alpha_i(t) = \alpha_i \exp\left(-\frac{(t - t_{i,0})^2}{2\sigma_{t,i}^2}\right). \tag{2}$$

Crucially, while Eq. 1 prescribes a linear trajectory, this assumption applies only to the center of each primitive within its valid temporal window, as regulated by the temporal opacity $\alpha_i(t)$. This temporal windowing mechanism limits each primitive's lifespan, enabling the system to approximate complex non-rigid motions through a piecewise linear representation. More complex deformations are captured through the sequential composition of short-lived dynamic primitives together with the continuous optimization of their appearance and geometry parameters. Finally, unlike offline approaches, the binary state $s_i$ is not fixed but is dynamically governed by the SA-KAN's uncertainty (see Sec. 3.5), enabling adaptive resource allocation.

### 3.2. Spatially-Aware Kolmogorov-Arnold Network

To identify dynamic regions without categorical priors, we propose the Spatially-Aware Kolmogorov-Arnold Network (SA-KAN), which maps DINOv2 features to a pixel-wise uncertainty map $\boldsymbol{\beta} \in \mathbb{R}_+^{H \times W}$. The architecture integrates depthwise separable convolutions for local context aggregation with a KAN (Liu et al., 2025), utilizing learnable B-Spline activations to precisely capture non-linear dynamic boundaries. This hybrid design is more parameter- and compute-efficient than the MLP baseline (details in Appendix A.2), with a final Softplus activation ensuring strictly positive values $\boldsymbol{\beta} > \mathbf{0}$.

### 3.3. The Optimization Paradox

To train the dual-stream representations, a naive approach would be to perform a joint static-dynamic optimization. While intuitively appealing, we identify a critical failure mode in this setup, termed the **Optimization Paradox**.

Consider a simplified joint formulation where the final rendered image $\hat{\mathbf{I}}_{joint}$ is a composition of static and dynamic streams weighted by the uncertainty map $\boldsymbol{\beta}$. From a probabilistic perspective, minimizing the Negative Log-Likelihood (NLL) with respect to the ground truth image $\mathbf{I}$ implies that the optimal uncertainty map $\boldsymbol{\beta}^*$ is element-wise proportional to the reconstruction residual magnitude $\mathcal{E}$:

$$\boldsymbol{\beta}^* \propto \|\mathcal{R}(\mathcal{G}_{joint}) - \mathbf{I}\| \approx \mathcal{E}, \tag{3}$$

where $\mathcal{R}(\cdot)$ denotes the differentiable rasterization operator. Ideally, $\boldsymbol{\beta}$ should be high only in dynamic regions where the static prior fails.

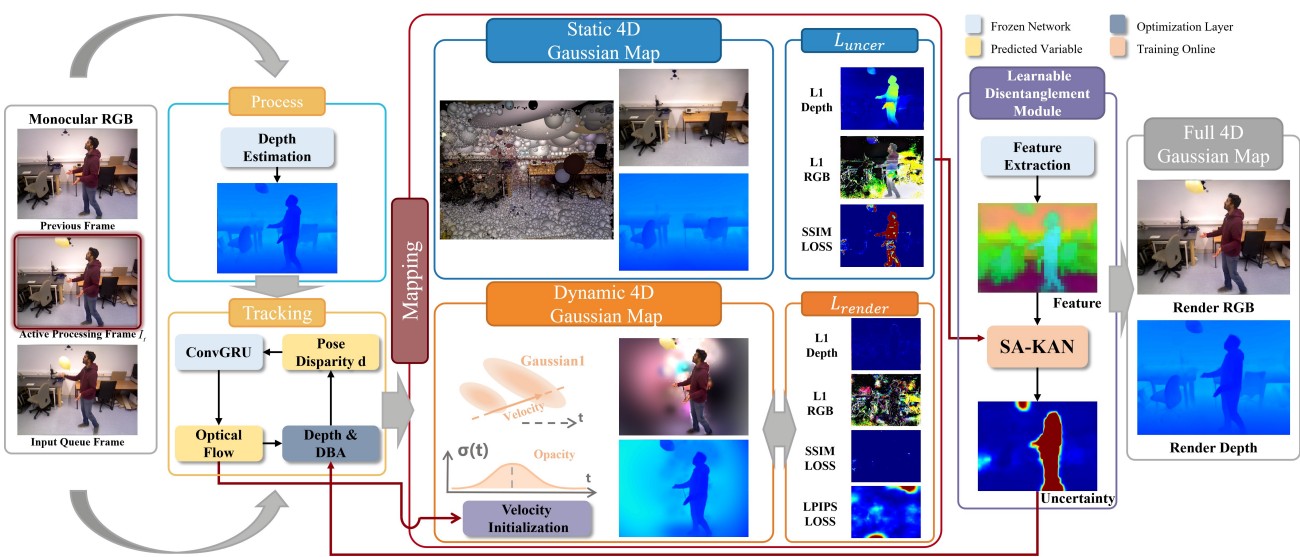

*Figure 2.* **System Overview of De4D-SLAM.** The pipeline takes monocular RGB frames as input and performs tracking and mapping in parallel through a decoupled dual-stream design. The tracking frontend estimates camera pose using optical flow, depth estimation, and dense bundle adjustment. The mapping backend maintains separate static and dynamic 4D Gaussian maps, while SA-KAN predicts a pixel-wise uncertainty map to activate dynamic primitives in high-uncertainty regions. Flow-Induced Initialization seeds new dynamic primitives, while gradient-isolated optimization prevents interference between the two streams during joint reconstruction. The final rendered RGB and depth outputs are produced by compositing the two streams.

However, the Capacity Mismatch between the two streams undermines this mechanism. Due to the high expressivity of 4D primitives, the dynamic stream $\mathcal{G}_d$ exhibits excessive plasticity, allowing it to rapidly overfit the entire scene, including static background details. Consequently, the joint reconstruction error $\mathcal{E}$ approaches zero globally before the SA-KAN converges. This leads to a phenomenon we term Uncertainty Collapse:

$$\lim_{\mathcal{E} \to \mathbf{0}} \boldsymbol{\beta}^* \to \mathbf{0}. \tag{4}$$

Since the dynamic stream effectively explains away all observations, the network is deceived into predicting low uncertainty ($\boldsymbol{\beta} \to \mathbf{0}$) everywhere. As shown in the joint visualization in Figure 1, the uncertainty map collapses into a trivial zero-solution (blue), failing to identify motion and consequently resulting in incomplete dynamic reconstruction with missing geometries.

### 3.4. Gradient-Isolated Optimization Strategy

To resolve the uncertainty collapse, we propose a gradient-isolated training scheme. Our core insight is to prevent the dynamic stream's overfitting from suppressing the supervisory signal for the uncertainty network. To formalize this, we introduce the stop-gradient operator sg[·]. While it acts as an identity mapping during the forward pass, it blocks error propagation during the backward pass:

$$\frac{\partial \mathrm{sg}[x]}{\partial x} = 0. \tag{5}$$

By utilizing sg[·], we mathematically enforce that $\boldsymbol{\beta}$ learns solely from the *fixed* static prior's failure, decoupling it from the dynamic stream's optimization.

**Static-Guided Uncertainty Learning.** We supervise the SA-KAN $\Phi_\theta$ utilizing the photometric inconsistency of the static reconstruction. We instantiate the abstract static residual $\mathcal{E}_s$ as a concrete structural loss $\mathcal{L}_{struct}$ between the rendered static image $\hat{\mathbf{I}}$ and the ground truth $\mathbf{I}$, balanced by hyperparameters $\lambda_s$ and $\lambda_d$:

$$\begin{aligned} \mathcal{L}_{struct}(\hat{\mathbf{I}}, \mathbf{I}) =& (1 - \lambda_s)\|\hat{\mathbf{I}} - \mathbf{I}\|_1 + \lambda_s \mathcal{L}_{ssim}(\hat{\mathbf{I}}, \mathbf{I}) \\ &+ \lambda_d \|\hat{\mathbf{D}} - \mathbf{D}_{metric}\|_1. \end{aligned} \tag{6}$$

Here, $\hat{\mathbf{D}}$ and $\mathbf{D}_{metric}$ denote the rendered static depth and the monocular metric depth prior, respectively.

To model the aleatoric uncertainty, we adopt an empirically stabilized regularization strategy inspired by the Laplacian distribution (see mathematical derivation in Appendix B.1). While the standard Laplacian NLL involves a $1/\beta$ term, we employ a Gaussian-like variance scaling to ensure numerical stability:

$$\mathcal{L}_{uncer} = \sum_{\mathbf{p}} \left( \frac{\mathrm{sg}[\mathcal{L}_{struct}(\hat{\mathbf{I}}_s, \mathbf{I})_{\mathbf{p}}]}{2\boldsymbol{\beta}(\mathbf{p})^2} + \frac{1}{2} \ln \boldsymbol{\beta}(\mathbf{p}) \right). \tag{7}$$

This hybrid formulation balances the robustness of $L_1$ residuals with the optimization stability of quadratic scaling.

Crucially, applying sg[·] to the residuals prevents the SA-KAN from altering the static geometry to reduce the loss. Since the static residual $\mathcal{L}_{struct}(\hat{\mathbf{I}}_s)$ remains high in dynamic regions (unlike the joint residual), this forces the corresponding values in $\boldsymbol{\beta}$ to increase, correctly highlighting moving objects.

**Uncertainty-Attenuated Reconstruction.** With dynamic regions identified, we perform a holistic optimization using a weighted objective:

$$\mathcal{L}_{render} = \sum_{\mathbf{p}} \omega(\mathbf{p}) \cdot \mathcal{L}_{struct}(\hat{\mathbf{I}}, \mathbf{I})_{\mathbf{p}} + \lambda_p \mathcal{L}_{lpips}(\hat{\mathbf{I}}, \mathbf{I}), \quad (8)$$

where $\lambda_p$ is the weight for the LPIPS perceptual loss. The adaptive weight $\omega(\mathbf{p})$ serves as a soft barrier, formulated as:

$$\omega(\mathbf{p}) = \text{sg}\left[\frac{1}{2\boldsymbol{\beta}(\mathbf{p})^2}\right]. \quad (9)$$

This detached weight ensures that the static stream focuses on reliable background regions (where $\boldsymbol{\beta}$ is low), while the dynamic stream is recruited to reconstruct outliers (where $\boldsymbol{\beta}$ is high), effectively resolving the optimization paradox.

### 3.5. Flow-Guided Dynamic Primitive Generation

To enable robust 4D reconstruction, we propose a generation mechanism that transitions primitives from a static prior to a dynamic state, utilizing flow-guided initialization to resolve the optimization cold start problem.

**Static Initialization and Triggering.** We tentatively initialize primitives in newly visible regions as static ($s_i = $ static, $\mathbf{v}_i = \mathbf{0}$), leveraging the scene's rigid prior. The activation of dynamic primitives is triggered when the SA-KAN consistently predicts high uncertainty $\boldsymbol{\beta}(\mathbf{p}) > \tau$, with the default value set to $\tau = 0.8$ based on the sensitivity analysis in Appendix Table 11.

**Flow-Guided Motion Initialization.** A critical challenge in 4DGS is that newly activated primitives often struggle to converge if initialized with zero velocity. To address this, we leverage dense optical flow $\mathbf{f}_{k \to k-1}$ alongside historical metric depth to formulate a physical motion prior.

Specifically, we lift the 2D optical flow into a 3D velocity vector. For an activated primitive at pixel $\mathbf{u}_i$ with depth $d_i$ in the current frame $k$, we first compute its corresponding 2D location in the previous frame as $\mathbf{u}'_i = \mathbf{u}_i + \mathbf{f}_{k \to k-1}(\mathbf{u}_i)$. Crucially, to recover motion along the optical axis, we explicitly reject the constant depth assumption. Instead, we sample the previous keyframe's metric depth map $D_{k-1}$ to obtain the conjugate depth $d'_i = D_{k-1}(\mathbf{u}'_i)$. The metric velocity $\mathbf{v}_{init}$ is then derived by differentiating the back-

projected 3D positions:

$$\begin{aligned} \mathbf{x}_k &= \mathbf{T}_k \pi^{-1}(\mathbf{u}_i, d_i), \\ \mathbf{x}_{k-1} &= \mathbf{T}_{k-1} \pi^{-1}(\mathbf{u}'_i, d'_i), \\ \mathbf{v}_{init} &= \frac{\mathbf{x}_k - \mathbf{x}_{k-1}}{\Delta t}, \end{aligned} \quad (10)$$

where $\mathbf{T}_k, \mathbf{T}_{k-1}$ denote the camera poses, and $\Delta t$ represents the time interval between frames. Based on this geometric prior, we initialize the canonical position and velocity of the dynamic primitive as $\mu_{i,0} = \mathbf{x}_k$ and $\mathbf{v}_i = \mathbf{v}_{init}$, respectively. By strictly anchoring $\mathbf{x}_{k-1}$ to the historical geometry $D_{k-1}$, this approach provides a robust physical prior for the non-convex 4D optimization.

### 3.6. Tracking and Mapping Integration

Our SA-KAN also serves as a critical feedback mechanism for the tracking frontend. Building on the DROID-SLAM architecture (Teed & Deng, 2021), we reformulate the Dense Bundle Adjustment (DBA) objective to account for both dynamic outliers and the inherent scale ambiguity of monocular video. The optimization minimizes the following energy function $E(\mathbf{T}, \mathbf{d})$:

$$E(\mathbf{T}, \mathbf{d}) = \sum_{(k,j) \in \mathcal{A}} \|\mathbf{r}_{kj}\|^2_{\mathbf{W}_{kj}} + \lambda_{geo} \sum_k \|\mathbf{d}_k - \mathbf{d}_{prior,k}\|_1. \quad (11)$$

where $\mathbf{T}$ represents the camera poses, $\mathbf{d}$ denotes the inverse depth map, and $\mathcal{A}$ is the set of edges in the frame graph. The term $\mathbf{r}_{kj}$ denotes the dense flow re-projection error between frame $k$ and $j$. To rigorously filter dynamic distractors, we introduce an uncertainty-modulated information matrix $\mathbf{W}_{kj}$:

$$\mathbf{W}_{kj}(\mathbf{p}) = (1 + \boldsymbol{\beta}_k(\mathbf{p}))^{-1} \cdot \mathbf{C}_{kj}(\mathbf{p})^{-1}, \quad (12)$$

where $\mathbf{C}_{kj}$ denotes the flow covariance. Crucially, our suppression factor $(1 + \boldsymbol{\beta}_k(\mathbf{p}))^{-1}$ naturally ensures that reliable static regions (where $\boldsymbol{\beta}_k(\mathbf{p}) \to 0$) retain full contribution weight ($\approx 1.0$). Conversely, for dynamic objects ($\boldsymbol{\beta}_k(\mathbf{p}) \gg 1$), the weight adaptively decays towards zero. Finally, the second term in Eq. 11, weighted by a coefficient $\lambda_{geo}$, anchors the estimated disparity $\mathbf{d}_k$ to a geometric prior $\mathbf{d}_{prior,k}$ derived from Metric3D (Hu et al., 2024).

## 4. Experiments

We evaluate the proposed De4D-SLAM framework on challenging benchmarks to validate its robustness and reconstruction fidelity. Our experiments are structured to assess three key dimensions: (1) **Tracking Robustness**: verifying localization accuracy in highly dynamic environments; (2)

*Table 1.* **Tracking Accuracy Comparison** (ATE RMSE [cm] ↓) on **Bonn Dataset**. "-" indicates unavailable results due to tracking divergence (**RTG-SLAM** on *Crowd*), runtime failure (**4DGS-SLAM** on *Rem.2*), or unreported data in the original source (**D4DGS-SLAM** (Sun et al., 2025)). Best results are highlighted in **green** (first) and yellow (second).

| Method | Balloon | Bal.2 | Person | Per.2 | Crowd | Crwd.2 | Sync | Sync2 | Rem. | Rem.2 | Place | Place2 | Avg. |
|---|---|---|---|---|---|---|---|---|---|---|---|---|---|
| RTG-SLAM [RGB-D] | 4.92 | 12.21 | 59.30 | 86.08 | - | 41.40 | 83.52 | 122.43 | 1.74 | 2.11 | 66.34 | 2.29 | 43.85 |
| DROID-SLAM [RGB] | 7.49 | 11.56 | 7.40 | 54.88 | 4.86 | 5.82 | 0.69 | 2.10 | 3.61 | 5.54 | 6.71 | 3.66 | 9.53 |
| DynaSLAM [RGB-D] | 3.00 | 2.90 | 6.10 | 7.80 | 1.60 | 3.10 | 23.20 | 3.90 | 4.50 | 4.80 | 5.20 | 4.00 | 5.84 |
| MonoGS [RGB] | 35.67 | 26.96 | 31.21 | 59.35 | 54.20 | 113.51 | 61.79 | 0.80 | 1.35 | 5.39 | 36.33 | 7.62 | 36.18 |
| SplaTAM [RGB-D] | 35.65 | 37.24 | 134.98 | 141.95 | 192.73 | 180.19 | 60.16 | 75.31 | 13.43 | 18.64 | 23.06 | 17.43 | 77.56 |
| WildGS-SLAM [RGB] | 2.74 | 2.82 | 3.45 | 3.08 | 1.67 | 1.91 | 0.72 | 1.00 | 1.43 | 1.85 | 1.62 | 1.97 | 2.02 |
| 4DGS-SLAM [RGB-D] | 2.50 | 4.20 | 8.70 | 8.76 | 2.17 | 74.68 | 3.45 | 0.54 | 1.23 | - | 33.56 | 1.54 | 12.84 |
| D4DGS-SLAM [RGB-D] | 3.60 | 3.90 | 4.50 | 5.20 | - | - | - | - | - | - | - | - | - |
| **Ours** [RGB] | 2.62 | 2.35 | 3.42 | 3.02 | 1.42 | 2.24 | 0.72 | 0.55 | 1.56 | 2.07 | 1.59 | 1.78 | 1.95 |

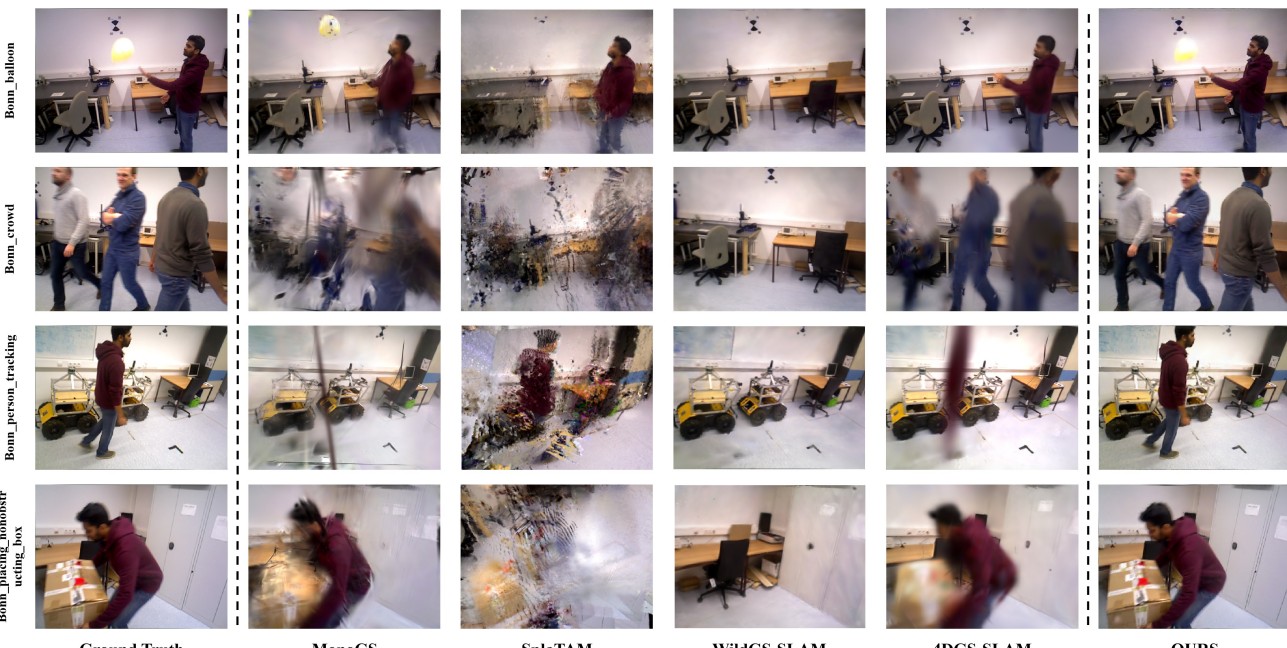

*Figure 3.* **Qualitative comparison of dynamic scene reconstruction. MonoGS** suffers from severe motion blur and ghosting. **WildGS** treats dynamic objects as outliers, resulting in missing geometry (black holes) or blurred inpainting. **De4D-SLAM (Ours)** successfully disentangles the scene, delivering photorealistic rendering of both static background and dynamic foreground agents.

**Holistic Reconstruction**: evaluating the visual fidelity of the reconstructed dynamic scenes; and (3) **Ablation Study**: analyzing core components, uncertainty predictor variants, and auxiliary priors to clarify their respective roles in the full system.

### 4.1. Experimental Setup

**Datasets.** We evaluate on two benchmarks: **TUM RGB-D** (Sturm et al., 2012), featuring controlled indoor human motion (e.g., `walking`), and **Bonn RGB-D** (Palazzolo et al., 2019), which captures more complex crowded scenes with intricate human-object interactions. These datasets provide diverse challenges ranging from simple foreground dynamics to frequent occlusions.

**Baselines.** We compare De4D-SLAM against a comprehensive set of state-of-the-art methods. First, for **General & Static SLAM**, we include DROID-SLAM (Teed & Deng, 2021) and 3DGS-based approaches such as MonoGS (Matsuki et al., 2024), SplaTAM (Keetha et al., 2024), and RTG-SLAM (Peng et al., 2024). These methods serve to demonstrate artifacts arising from geometric violations in dynamic scenes. Second, we evaluate the **Robust Mapping (Exclusion-based)** paradigm using DynaSLAM (Bescos et al., 2018) and WildGS-SLAM (Zheng et al., 2025). These approaches treat dynamic objects purely as outliers to be masked. Finally, we benchmark against concurrent **Dynamic 4D SLAM** frameworks, specifically 4DGS-SLAM (Li et al., 2025) and D4DGS-SLAM (Sun et al., 2025). In Tables 1, 2, and 3, we annotate each method with

*Table 2.* **Reconstruction Quality Comparison on Bonn Dataset.** We report per-sequence **PSNR** [dB] ↑ and overall averages. *WildGS masks out dynamic regions, thus cannot render the full scene (N/A). "-" indicates measurement unavailable due to runtime failure (**4DGS-SLAM** on *Rem.2*) or due to incomplete sequence-level reporting in the original source, which prevents directly comparable full-split averages (**D4DGS-SLAM** (Sun et al., 2025)). Best results are in bold.

| Method | PSNR ↑ (Per Sequence) | | | | | | | | | | | | Average | | |
| | Balloon | Bal.2 | Person | Per.2 | Crowd | Crwd.2 | Sync | Sync2 | Rem. | Rem.2 | Place | Place2 | PSNR↑ | SSIM↑ | LPIPS↓ |
|---|---|---|---|---|---|---|---|---|---|---|---|---|---|---|---|
| MonoGS [RGB] | 20.91 | 19.23 | 19.22 | 20.52 | 16.91 | 19.95 | 21.45 | 19.93 | 22.15 | 21.52 | 18.91 | 20.96 | 20.14 | 0.760 | 0.360 |
| SplaTAM [RGB-D] | 19.54 | 17.84 | 16.29 | 16.15 | 13.68 | 15.12 | 19.29 | 19.30 | 21.66 | 21.26 | 20.04 | 21.63 | 18.48 | 0.722 | 0.267 |
| WildGS-SLAM* [RGB] | N/A | N/A | N/A | N/A | N/A | N/A | N/A | N/A | N/A | N/A | N/A | N/A | N/A | N/A | N/A |
| 4DGS-SLAM [RGB-D] | 26.23 | 23.10 | 21.08 | 19.78 | 22.17 | 17.01 | 23.89 | **25.38** | 23.96 | - | 21.14 | 24.83 | 22.60 | 0.833 | 0.279 |
| D4DGS-SLAM [RGB-D] | **27.89** | **29.65** | **27.66** | **31.18** | - | - | - | - | - | - | - | - | - | - | - |
| **Ours** [RGB] | 27.31 | 28.43 | 27.49 | 27.15 | **27.49** | **28.78** | **27.88** | 23.28 | **28.66** | **22.79** | **25.24** | **25.89** | 26.70 | **0.903** | **0.200** |

*Table 3.* **Tracking Accuracy Comparison** (ATE RMSE [cm] ↓) on **TUM RGB-D Dataset**. We report results on dynamic *fr3* sequences. Best results are highlighted in **green** (first) and yellow (second).

| Method | fr3_ss | fr3_sx | fr3_ws | fr3_wx | fr3_wr | fr3_wh | Avg. |
|---|---|---|---|---|---|---|---|
| RTG-SLAM [RGB-D] | 0.69 | 0.88 | 1.23 | 29.45 | 58.86 | 31.37 | 20.41 |
| DROID-SLAM [RGB] | 0.63 | 1.79 | 0.69 | 2.04 | 5.90 | 4.28 | 2.56 |
| DynaSLAM [RGB-D] | 0.50 | 1.50 | 0.60 | 1.50 | 3.50 | 2.50 | 1.68 |
| MonoGS [RGB] | 0.90 | 2.28 | 8.74 | 21.50 | 17.40 | 44.20 | 15.84 |
| SplaTAM [RGB-D] | 0.50 | 0.90 | 2.30 | 1.30 | 3.90 | 2.20 | 1.85 |
| WildGS-SLAM [RGB] | 0.50 | 0.82 | 0.40 | 1.30 | 3.30 | 1.60 | 1.32 |
| 4DGS-SLAM [RGB-D] | 0.58 | 2.90 | 0.52 | 2.10 | 2.60 | 2.42 | 1.85 |
| **Ours** [RGB] | 0.50 | 0.83 | 0.49 | **1.20** | 3.20 | **1.40** | **1.27** |

its input modality, where [RGB] denotes monocular RGB input and [RGB-D] denotes RGB-D input.

**Evaluation Metrics.** For tracking, we report the Absolute Trajectory Error (ATE) RMSE [cm] after Sim(3) alignment to correct for scale ambiguity. Reconstruction fidelity is evaluated on all keyframes using PSNR, SSIM, and LPIPS (Zhang et al., 2018).

### 4.2. Comparisons

**Tracking Performance in Dynamic Environments.** As summarized in Table 1 and Table 3, De4D-SLAM achieves competitive localization accuracy across both the interaction-heavy Bonn benchmark and the human-centric TUM RGB-D benchmark. On Bonn, our method obtains the lowest average ATE of 1.95 cm among the evaluated methods, while substantially reducing drift relative to static-scene-oriented baselines such as MonoGS and SplaTAM in highly dynamic sequences. At the same time, De4D-SLAM remains comparable to masking-based methods such as WildGS in tracking robustness, indicating that our unified framework preserves reliable geometric constraints for pose estimation under dynamic motion. Compared with concurrent 4DGS frameworks such as D4DGS-SLAM (Sun et al., 2025), which rely on external dynamics-aware modules, our method also maintains competitive tracking performance in a purely self-supervised setting. On TUM RGB-D, De4D-SLAM reduces the tracking error of the DROID backbone by 50.4%, showing that the proposed Gradient-Isolated strategy improves the quality of the geometric constraints used

for optimization under rapid non-rigid motion.

**Holistic 4D Reconstruction Quality.** Quantitative metrics in Table 2 and visual evidence in Fig. 3 highlight the efficacy of our dual-stream architecture in resolving the Optimization Paradox. While masking-based methods such as WildGS can maintain competitive tracking by rejecting dynamic regions as outliers, this strategy removes dynamic content from the final map. By contrast, De4D-SLAM preserves competitive localization performance while jointly reconstructing static background and dynamic foreground in a unified representation. Existing methods typically face a dilemma between incomplete voids (WildGS) and over-smoothed blur (4DGS-SLAM). The latter's blurriness is quantitatively reflected in its higher LPIPS (0.279) compared to ours (0.200). This artifact arises because 4DGS-SLAM relies on coarse categorical priors that lack the granularity needed to supervise fine-grained deformation, causing dynamic primitives to settle into temporally averaged geometry. De4D-SLAM alleviates this issue by recovering sharper high-frequency details and more coherent scene geometry. While D4DGS-SLAM reports strong PSNR on some of the available sequences, such as Per.2, the incomplete availability of its sequence-level results limits a comprehensive comparison across all settings.

**Overall Comparison.** Taken together, these comparisons indicate that De4D-SLAM addresses a broader objective than exclusion-based dynamic SLAM methods. While remaining competitive in localization performance, it reconstructs both static background and dynamic foreground within a unified representation, yielding a more complete and coherent dynamic scene model. We further validate this generalization behavior on the full WildGS MoCap split (Zheng et al., 2025) in Appendix Table 14, where De4D-SLAM remains comparable to WildGS-SLAM in tracking accuracy. We also provide qualitative examples on the WildGS iPhone split (Zheng et al., 2025) in Appendix Fig. 6, which contains mixed indoor and outdoor casual captures without trajectory ground truth.

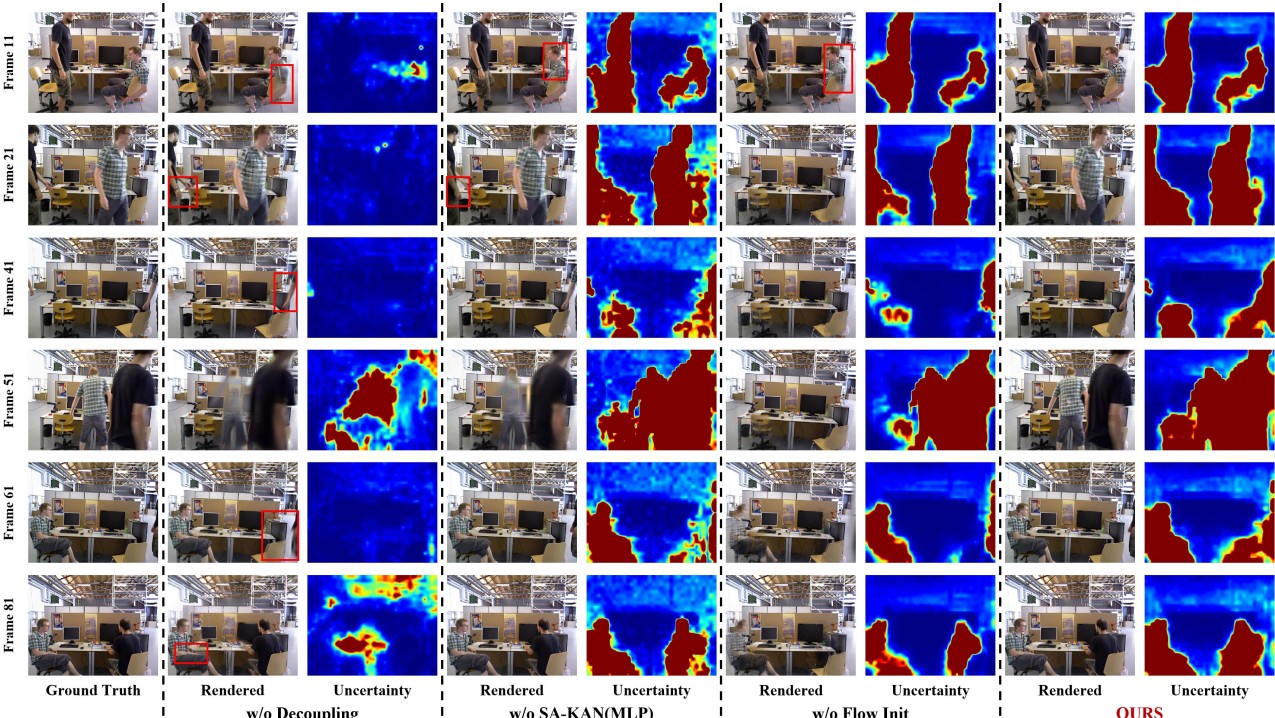

*Figure 4*. **Visual Ablation on the *walking_static* Sequence.** We compare the rendered RGB and learned uncertainty masks $\beta$ across different variants over time. **w/o Decoupling:** The naive joint optimization suffers from mask collapse (masks appear uniformly static), leading to ghosting artifacts in dynamic regions. **w/o SA-KAN (MLP):** The lack of spatial aggregation results in noisy, salt-and-pepper uncertainty masks, causing false positives on the static background. **w/o Flow Init:** Fails to capture sharp motion boundaries due to convergence into local minima. **Ours:** Successfully disentangles the scene with spatially coherent masks, delivering artifact-free rendering and precise motion segmentation.

## 4.3. Ablation Study

We study the contribution of the major modules in De4D-SLAM through complementary ablation studies. We first analyze the core architectural components on the TUM RGB-D benchmark, which provides a controlled setting for evaluating decoupling, uncertainty prediction, and flow-guided initialization. We then report supplementary ablations on the more interaction-heavy Bonn benchmark to assess the robustness of the uncertainty predictor and the contribution of auxiliary priors under more challenging dynamic scenes.

**Core Component Ablations.** To quantify the contribution of each core module, we evaluate three variants on the TUM RGB-D dynamic sequences: **w/o Decoupling**, which removes the gradient-isolated training scheme; **w/o SA-KAN**, which replaces the uncertainty predictor with a standard pixel-wise MLP; and **w/o Flow Init**, which disables velocity initialization for newly activated dynamic primitives. Quantitative and visual results are reported in Table 4 and Fig. 4. Removing gradient-isolated decoupling leads to the largest overall degradation, confirming the effectiveness of the proposed gradient-isolated dual-stream framework in preventing uncertainty collapse. Replacing SA-KAN with

a standard MLP preserves basic functionality but weakens the spatial coherence of the uncertainty prediction, which in turn reduces the reliability of the geometric constraints used by the tracking frontend. Disabling flow-guided initialization has a comparatively limited effect on tracking accuracy but substantially degrades reconstruction fidelity, indicating that a suitable warm start is particularly important for modeling fast dynamic motion.

*Table 4*. **Core Component Ablations.** We report average tracking accuracy (**ATE** [cm] ↓) and reconstruction quality (**PSNR** [dB] ↑, **SSIM** ↑, **LPIPS** ↓) across TUM dynamic sequences. Best results are in **bold**.

| Variant | Tracking ATE ↓ | Rendering Quality PSNR ↑ | SSIM ↑ | LPIPS ↓ |
|---|---|---|---|---|
| w/o Decoupling | 1.31 | 23.10 | 0.817 | 0.242 |
| w/o Flow Init | 1.28 | 20.33 | 0.772 | 0.262 |
| *Architecture:* | | | | |
| w/o SA-KAN (MLP) | 1.37 | 26.73 | 0.897 | 0.176 |
| w/o Spatial Agg. | 1.38 | 26.55 | 0.883 | 0.186 |
| **Ours** | **1.27** | **27.59** | **0.911** | **0.164** |

*Table 5.* **Uncertainty Predictor Variants.** Average tracking accuracy and reconstruction quality on the Bonn benchmark. We compare a standard pixel-wise MLP, a spatially augmented MLP (SA-MLP), and the proposed SA-KAN. Best results are in **bold**.

| Predictor | ATE ↓ | PSNR ↑ | SSIM ↑ | LPIPS ↓ |
|---|---|---|---|---|
| MLP | 2.01 | 26.68 | 0.900 | 0.210 |
| SA-MLP | 2.04 | **26.96** | **0.907** | 0.211 |
| SA-KAN | **1.95** | 26.70 | 0.903 | **0.200** |

*Table 6.* **Auxiliary Prior Ablations.** Average tracking accuracy and reconstruction quality on the Bonn benchmark. We evaluate the contribution of the Metric3D geometric prior and DINOv2 features in the full system. Best results are in **bold**.

| Variant | ATE ↓ | PSNR ↑ | SSIM ↑ | LPIPS ↓ |
|---|---|---|---|---|
| w/o Metric3D | 2.42 | 24.73 | 0.864 | 0.266 |
| w/o DINOv2 | 2.21 | **27.23** | **0.912** | 0.214 |
| Full Model | **1.95** | 26.70 | 0.903 | **0.200** |

**Uncertainty Predictor Variants.** We further compare three uncertainty predictors on the Bonn benchmark: a standard pixel-wise MLP, a spatially augmented MLP (SA-MLP), and the proposed SA-KAN. This comparison isolates the respective roles of spatial aggregation and the KAN-based predictor under more challenging dynamic interactions. The quantitative results are summarized in Table 5. The results show that spatial context is beneficial, as both SA-MLP and SA-KAN improve over the plain MLP in reconstruction quality. Although SA-MLP achieves slightly higher PSNR and SSIM, SA-KAN attains the best ATE and LPIPS, indicating a better trade-off between reconstruction fidelity and localization robustness. This pattern suggests that augmenting the predictor with spatial context already improves uncertainty coherence, while the KAN-based design is particularly beneficial for preserving cleaner uncertainty boundaries and more stable pose estimation. Since the uncertainty map directly modulates which image regions are trusted by the tracking frontend, its quality is evaluated here indirectly through downstream localization and reconstruction behavior rather than through a separate segmentation benchmark.

**Auxiliary Prior Ablations.** We study the contribution of the auxiliary priors used in the full system on the Bonn benchmark by removing the Metric3D geometric prior and by replacing DINOv2 features with shallow RGB convolutional features. The corresponding results are reported in Table 6. These ablations show that the two priors contribute in different ways to the final system behavior. Metric3D primarily improves geometric scale stability and tracking robustness. By contrast, although the shallow RGB variant slightly improves PSNR and SSIM, the full model still achieves the best overall balance in terms of ATE and LPIPS.

*Table 7.* **Runtime Breakdown.** Average online runtime of the major components on the Bonn *Crowd* sequence.

| Module | Avg. runtime |
|---|---|
| SA-KAN inference | 2.57 ms/call |
| DINOv2 feature extraction | 7.1 ms/frame |
| Optical flow computation | 13.3 ms/frame |
| Local DBA pose optimization | 517.8 ms/frame |
| Backend dynamic-static Gaussian optimization | 1224.6 ms/frame |
| Full system | 2.14 s/frame (0.47 FPS) |

This suggests that DINOv2 provides richer visual cues for uncertainty prediction, which are particularly beneficial for localization reliability and perceptual fidelity.

**4.4. Runtime Analysis**

Runtime profiling on the Bonn *Crowd* sequence shows that the full system runs at 0.47 FPS (2.14 s/frame). While this implementation is not yet real-time, it follows the typical efficiency pattern of Gaussian-Splatting-based SLAM systems. Once dense dynamic scene modeling is introduced, the computational budget is dominated by Gaussian optimization rather than by feature extraction or uncertainty prediction. In our system, DINOv2 feature extraction, optical flow computation, and SA-KAN inference account for only a very limited portion of the total cost. Overall, this runtime behavior is consistent with that of unified Gaussian-Splatting-based pipelines that jointly support tracking and holistic dynamic reconstruction.

# 5. Conclusion

In this work, we presented **De4D-SLAM**, a unified framework for dense monocular SLAM in dynamic environments. By introducing a Gradient-Isolated Dual-Stream Architecture and the Spatially-Aware KAN, we effectively resolve the optimization paradox inherent in class-agnostic dynamic reconstruction. Our method successfully decouples static background from dynamic entities without relying on predefined semantic categories. Extensive experiments demonstrate that De4D-SLAM achieves competitive performance in both tracking accuracy and rendering fidelity, offering a robust perception solution for embodied agents operating in complex, time-varying scenes. These results indicate that explicit static-dynamic decoupling can preserve localization robustness while enabling richer dynamic reconstruction in monocular SLAM. Ultimately, this framework expands dynamic SLAM from a narrow trajectory-estimation tool to a holistic 4D spatial modeling engine, with limitations and future work detailed in Appendix E.

## Acknowledgements

This work was supported in part by the "Scientists + Engineers" project of Qin Chuangyuan in Shaanxi Province of China under Grant 2024QCY-KXJ-152, the Key Industry Chain Technology Research and Development General Project of Xi'an under Grant 2024JH-CLYB-0047, the National Natural Science Foundation of China under Grants 62371373, 62271377, 62401418, and 62501450, the China Postdoctoral Science Foundation under Grant 2024M762544, and the Fundamental Research Funds for the Central Universities.

## Impact Statement

This paper presents advancements in dynamic SLAM and 4D scene reconstruction, with the goal of enabling embodied agents to operate robustly in unpredictable, open-world environments. By removing the reliance on pre-defined semantic categories, our work significantly enhances the safety and adaptability of autonomous systems (e.g., service robots, autonomous vehicles) in scenarios involving unknown moving objects, potentially reducing accident risks in unstructured settings.

However, we acknowledge that the capability to reconstruct high-fidelity 4D representations of dynamic agents, specifically humans, carries potential privacy implications. The precise decoupling of dynamic entities could be repurposed for unauthorized surveillance or the creation of digital twins without consent. While our current focus is on geometric reconstruction for navigation, we strongly advocate for the integration of privacy-preserving mechanisms, such as real-time blurring or feature de-identification, in future deployments of such technology. We believe that the benefits of robust perception in safety-critical applications outweigh these risks, provided that strict data governance and ethical deployment standards are maintained.

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

# A. System Implementation Details

## A.1. Full System Workflow

Our framework orchestrates motion-aware preprocessing, uncertainty-guided tracking, and gradient-isolated mapping into a cohesive and efficient pipeline. The detailed workflow is delineated below.

**Motion-Filtered Pre-processing.** To optimize computational resource allocation, incoming video frames undergo a motion filtering process based on the mean magnitude of the optical flow. Frames exhibiting sufficient parallax, specifically where the mean optical flow magnitude exceeds $\tau_{motion} = 3.0$ pixels, or those satisfying a maximum temporal interval of $N_{max} = 9$ frames, are selected as keyframes. This strategy effectively filters out redundant static frames, ensuring that the 4D Gaussian map is populated exclusively with geometrically meaningful observations.

**Uncertainty-Aware Tracking Frontend.** The frontend leverages a recurrent optical flow network, built upon the DROID-SLAM architecture (Teed & Deng, 2021), for dense correspondence and camera pose estimation. A pivotal enhancement in our implementation is the augmentation of the Dense Bundle Adjustment (DBA) layer. Unlike standard implementations that assume a uniform noise distribution, our DBA layer actively integrates uncertainty feedback from the mapping thread. Specifically, the global uncertainty map $\beta$, maintained by the backend, is projected into the current camera view using the predicted pose. This projected uncertainty modulates the information matrix $\mathbf{W}_{kj}$ within the re-projection error minimization objective. Consistent with Eq. 12 in the main text, we define the weights as $\mathbf{W}_{kj}(\mathbf{p}) = (1 + \boldsymbol{\beta}_k(\mathbf{p}))^{-1} \cdot \mathbf{C}_{kj}(\mathbf{p})^{-1}$. By dynamically down-weighting contributions from regions with high predicted uncertainty, the system effectively shields the camera pose optimization from corruption by dynamic outliers.

**Gradient-Isolated Mapping Backend.** The backend operates in a self-supervised regime to maintain the global 4D Gaussian map. For every new keyframe, the system extracts high-level features using DINOv2 (Oquab et al., 2024) and feeds them into the proposed Spatially-Aware KAN (SA-KAN) to predict a pixel-wise uncertainty map $\boldsymbol{\beta}$. This map orchestrates the *Dual-Stream Primitive Management*: regions where $\boldsymbol{\beta}(\mathbf{p}) < \tau$ initialize static Gaussian primitives, while regions where $\boldsymbol{\beta}(\mathbf{p}) > \tau$ trigger the initialization of dynamic primitives. To address the inherent non-convexity of dynamic reconstruction, we implement the **Flow-Induced Initialization** strategy (detailed in Sec. 3.5), lifting 2D optical flow fields into 3D velocity vectors to provide a physical warm start. During the online optimization phase, we employ a **Gradient-Isolated** loss function. This mechanism strictly blocks the gradients of the uncertainty-regularized loss term from propagating into the static geometry, thereby preventing the optimization paradox described in Sec. 3.3, where dynamic priors might erroneously degrade the static reconstruction.

*Table 8.* **Complexity Analysis.** Comparison of operational costs between the MLP baseline and our SA-KAN on a $32 \times 32$ feature grid. Best results are in bold.

| Architecture | Params (k) ↓ | FLOPs (M) ↓ | Time (ms) ↓ |
|---|---|---|---|
| MLP Baseline | 28.86 | 29.43 | **0.30** |
| **SA-KAN (Ours)** | **10.75** | **11.40** | 2.57 |

## A.2. Network Architecture: SA-KAN

The Spatially-Aware Kolmogorov-Arnold Network (SA-KAN) serves as the core of the Learnable Disentanglement Module (LDM). The architecture comprises two primary stages designed for high efficiency.

**Stage 1: Spatial Aggregation Module.** This module injects a geometric inductive bias into the visual foundation features. It processes the input feature map ($\mathbf{F}_{in} \in \mathbb{R}^{H/14 \times W/14 \times 384}$) using a depthwise separable convolution with a $5 \times 5$ kernel, padding of 2, and stride of 1. A residual connection preserves the original feature identity:

$$\mathbf{F}_{spatial} = \mathbf{F}_{in} + \text{DWConv}_{5 \times 5}(\mathbf{F}_{in})$$

**Stage 2: KAN Projection Head.** The aggregated features are mapped to a scalar uncertainty value via a two-layer Linear KAN (Liu et al., 2025). The first layer maps the 384-channel input to a 64-channel latent space, followed by a second layer mapping to a 1-channel output. Both layers utilize a grid size of $G = 3$ and a spline order of $k = 3$, with SiLU as the base activation. The final output employs a Softplus activation to strictly constrain the uncertainty to the positive domain ($\boldsymbol{\beta} > \mathbf{0}$).

**Complexity Analysis.** To assess operational efficiency, we benchmark SA-KAN against a standard MLP baseline (3 layers, 64 hidden units) on a $32 \times 32$ feature grid using a single NVIDIA Tesla V100 GPU (32GB). As shown in Table 8, the SA-KAN configuration used in our system employs **62.8%** fewer parameters (10.75k vs. 28.86k) and **61.3%** fewer FLOPs (11.40 M vs. 29.43 M) than the compared MLP baseline. Although the B-spline calculation introduces a slight latency increase (2.57 ms), this cost remains small relative to the mapping thread's overall budget ($>$50 ms) and therefore does not constitute a practical bottleneck in our implementation.

*Table 9.* **Hyperparameter Configuration.** We adopt standard Gaussian Splatting settings augmented with our dynamic-specific parameters.

| Module | Parameter | Value |
|---|---|---|
| Gaussian Optimization | Position Learning Rate | $1.6 \times 10^{-4} \to 1.6 \times 10^{-6}$ (Log Decay) |
| | Feature (SH) LR | $2.5 \times 10^{-3}$ |
| | Opacity LR | 0.05 |
| | Scaling LR | 0.005 |
| | Rotation LR | 0.001 |
| Loss Weights | RGB ($\lambda_1$) | $1.0 \times (1 - \lambda_{ssim})$ |
| | SSIM ($\lambda_{ssim}$) | 0.2 |
| | Depth ($\lambda_{depth}$) | 0.1 |
| | Uncertainty ($\lambda_{unc}$) | Adaptive (Eq. 9) |
| System Thresholds | Dynamic Trigger ($\tau$) | 0.8 |
| | Keyframe Flow Thresh. ($\tau_{motion}$) | 3.0 px |
| | Max Keyframe Interval ($N_{max}$) | 9 frames |
| | Mapping Iterations | 60 per keyframe |
| | BA Frequency | Every 20 keyframes |

*Table 10.* **Primitive Count Analysis on Bonn *Crowd*.** Statistics of dynamic primitives under adaptive density control.

| Quantity | Value |
|---|---|
| New short-lived dynamic primitives / keyframe | ~2,389 |
| Recycled / deactivated primitives / keyframe | ~425.8 |
| Active short-lived primitive buffer | ~4,743 |
| Peak dynamic primitive count | 215,732 |
| Peak dynamic ratio in full scene capacity | 14.27% |

*Table 11.* **Sensitivity to Dynamic Trigger Threshold $\tau$.** Results on the Bonn *Crowd* sequence. Best results are in **bold**.

| $\tau$ | ATE $\downarrow$ | PSNR $\uparrow$ |
|---|---|---|
| 0.3 | 1.51 | 26.68 |
| 0.5 | 1.52 | 27.31 |
| 0.8 | **1.42** | **27.49** |
| 1.0 | 1.49 | 27.10 |

### A.3. Adaptive Density Control for Dynamic Primitives

Unlike static scenes where geometry is temporally consistent, dynamic entities undergo rapid topological changes (e.g., occlusion, disocclusion). Standard density control strategies (Kerbl et al., 2023) often fail to adapt to these shifts, leading to floater artifacts in empty space. To address this, we implement a **Motion-Aware Pruning** strategy specifically for the dynamic stream $\mathcal{G}_d$.

**Velocity-Based Pruning.** We observe that outliers in the dynamic stream often exhibit physically implausible velocities. We periodically prune primitives whose predicted velocity magnitude $\|\mathbf{v}_i\|$ exceeds a threshold $\tau_{vel} = 5.0$ m/s (assuming indoor scale) or whose trajectory projects outside the camera frustum for more than $k = 5$ consecutive frames.

**Uncertainty-Guided Densification.** Standard densification relies on view-space positional gradients. We augment this by incorporating the uncertainty signal. Primitives are candidates for cloning not only if their positional gradient is high but also if they reside in regions of high epistemic uncertainty ($\boldsymbol{\beta}(\mathbf{p}) > 0.9$). This actively encourages the system to allocate more Gaussian primitives to represent complex, rapidly moving boundaries, ensuring high-fidelity reconstruction of non-rigid deformations.

### A.4. Hardware and Hyperparameters

All experiments were conducted on a workstation equipped with a single NVIDIA Tesla V100 GPU (32GB). The system is implemented in PyTorch with customized CUDA rasterization kernels. Key hyperparameters used in our evaluation are listed in Table 9.

To examine the sensitivity of the dynamic trigger threshold $\tau$, we evaluate several values on the Bonn *Crowd* sequence. The results in Table 11 show that the method remains stable across a practical range of settings. Among the tested values, $\tau = 0.8$ provides the best overall trade-off between tracking accuracy and reconstruction quality.

## B. Mathematical Derivations

### B.1. Uncertainty Loss: Probabilistic Basis and Stabilization

In the main text (Sec. 3.4), we introduced the uncertainty-aware objective $\mathcal{L}_{uncer}$. Here, we detail the theoretical motivation rooted in the Laplacian distribution and justify our empirically stabilized implementation.

**Theoretical Motivation.** Since our structural loss $\mathcal{L}_{struct}$ is dominated by $L_1$ terms, we model the residual $r(\mathbf{p})$ at pixel $\mathbf{p}$ as a random variable following a **Laplacian distri-**

*Table 12.* **Reconstruction Quality Comparison on TUM RGB-D Dataset.** We report **PSNR** [dB] ↑, **SSIM** ↑, and **LPIPS** ↓. **De4D-SLAM** achieves the best average performance across the three reconstruction metrics.

| Method | Metric | sitting_static | sitting_xyz | walking_static | walking_xyz | walking_rpy | walking_halfsphere | Avg. |
|---|---|---|---|---|---|---|---|---|
| MonoGS (Matsuki et al., 2024) | PSNR ↑ | 21.44 | 21.86 | 13.87 | 12.69 | 14.62 | 13.80 | 16.38 |
| | SSIM ↑ | 0.792 | 0.799 | 0.510 | 0.349 | 0.492 | 0.450 | 0.565 |
| | LPIPS ↓ | 0.159 | 0.177 | 0.384 | 0.564 | 0.555 | 0.596 | 0.406 |
| SplaTAM (Keetha et al., 2024) | PSNR ↑ | 24.35 | 21.83 | 19.71 | 17.26 | 16.26 | 14.89 | 19.05 |
| | SSIM ↑ | 0.919 | **0.878** | 0.773 | 0.658 | 0.627 | 0.528 | 0.731 |
| | LPIPS ↓ | **0.100** | **0.165** | 0.223 | 0.338 | 0.355 | 0.439 | 0.270 |
| 4DGS-SLAM (Li et al., 2025) | PSNR ↑ | 27.68 | 24.37 | 22.99 | 19.83 | 19.22 | 19.67 | 22.29 |
| | SSIM ↑ | 0.892 | 0.822 | 0.820 | 0.730 | 0.708 | 0.717 | 0.782 |
| | LPIPS ↓ | 0.116 | 0.179 | 0.195 | 0.281 | 0.337 | 0.325 | 0.239 |
| **Ours** | PSNR ↑ | **28.64** | **25.14** | **27.84** | **27.11** | **28.78** | **28.03** | **27.59** |
| | SSIM ↑ | **0.924** | 0.877 | **0.915** | **0.912** | **0.917** | **0.919** | **0.911** |
| | LPIPS ↓ | 0.116 | 0.183 | **0.128** | **0.173** | **0.193** | **0.192** | **0.164** |

*Table 13.* **Reconstruction Quality Comparison on Bonn RGB-D Dataset.** WildGS (Zheng et al., 2025) is omitted as it masks out dynamic regions. **De4D-SLAM** achieves strong reconstruction fidelity.

| Method | Metric | Balloon | Bal.2 | Person | Per.2 | Crowd | Crwd.2 | Sync | Sync2 | Rem. | Rem.2 | Place | Place2 | Avg. |
|---|---|---|---|---|---|---|---|---|---|---|---|---|---|---|
| MonoGS | PSNR ↑ | 20.91 | 19.23 | 19.22 | 20.52 | 16.91 | 19.95 | 21.45 | 19.93 | 22.15 | 21.52 | 18.91 | 20.96 | 20.14 |
| | SSIM ↑ | 0.786 | 0.745 | 0.742 | 0.764 | 0.656 | 0.661 | 0.749 | 0.822 | 0.839 | **0.837** | 0.718 | 0.806 | 0.760 |
| | LPIPS ↓ | 0.334 | 0.361 | 0.393 | 0.349 | 0.499 | 0.506 | 0.321 | 0.237 | 0.281 | 0.286 | 0.440 | 0.317 | 0.360 |
| SplaTAM | PSNR ↑ | 19.54 | 17.84 | 16.29 | 16.15 | 13.68 | 15.12 | 19.29 | 19.30 | 21.66 | 21.26 | 20.04 | 21.63 | 18.48 |
| | SSIM ↑ | 0.781 | 0.707 | 0.643 | 0.599 | 0.509 | 0.585 | 0.776 | 0.719 | 0.860 | 0.854 | 0.785 | 0.850 | 0.722 |
| | LPIPS ↓ | 0.221 | 0.286 | 0.326 | 0.351 | 0.407 | 0.363 | 0.235 | 0.277 | **0.152** | **0.181** | 0.221 | 0.182 | 0.267 |
| 4DGS-SLAM | PSNR ↑ | 26.23 | 23.10 | 21.08 | 19.78 | 22.17 | 17.01 | 23.89 | **25.38** | 23.96 | - | 21.14 | 24.83 | 22.60 |
| | SSIM ↑ | 0.874 | 0.841 | 0.828 | 0.812 | 0.820 | 0.729 | 0.810 | **0.890** | 0.873 | - | 0.810 | 0.876 | 0.833 |
| | LPIPS ↓ | 0.237 | 0.268 | 0.296 | 0.306 | 0.313 | 0.443 | 0.248 | **0.175** | 0.254 | - | 0.310 | 0.219 | 0.279 |
| **Ours** | PSNR ↑ | **27.31** | **28.43** | **27.49** | **27.15** | **27.49** | **28.78** | **27.88** | 23.28 | **28.66** | **22.79** | **25.24** | **25.89** | **26.70** |
| | SSIM ↑ | **0.929** | **0.934** | **0.929** | **0.933** | **0.921** | **0.930** | **0.919** | 0.822 | **0.934** | 0.795 | **0.879** | **0.914** | **0.903** |
| | LPIPS ↓ | **0.173** | **0.166** | **0.191** | **0.174** | **0.193** | **0.192** | **0.198** | 0.273 | 0.191 | **0.198** | 0.245 | 0.209 | **0.200** |

**bution**. Assuming a heteroscedastic scale parameter $\beta(\mathbf{p})$, the probability density function is:

$$p(r|\boldsymbol{\beta}) = \frac{1}{2\boldsymbol{\beta}(\mathbf{p})} \exp\left(-\frac{|r(\mathbf{p})|}{\boldsymbol{\beta}(\mathbf{p})}\right). \qquad (13)$$

Maximizing the log-likelihood is equivalent to minimizing the Negative Log-Likelihood (NLL). Through logarithmic transformation, we derive:

$$\mathcal{L}_{NLL} = -\ln p(r|\boldsymbol{\beta}) \propto \ln \boldsymbol{\beta}(\mathbf{p}) + \frac{|r(\mathbf{p})|}{\boldsymbol{\beta}(\mathbf{p})}. \qquad (14)$$

Dropping the additive constant $\ln(2)$, the canonical optimization objective becomes:

$$\mathcal{L}_{canonical} = \frac{|r(\mathbf{p})|}{\boldsymbol{\beta}(\mathbf{p})} + \ln \boldsymbol{\beta}(\mathbf{p}). \qquad (15)$$

**Empirical Stabilization.** While Eq. 15 is theoretically sound, directly optimizing the linear denominator $1/\boldsymbol{\beta}(\mathbf{p})$ often leads to gradient instability when the predicted uncertainty approaches zero. To address this, we adopt a hybrid formulation. We approximate the denominator using a Gaussian-like variance scaling $(2\boldsymbol{\beta}(\mathbf{p})^2)$ while retaining

the robust $L_1$ residual ($|r| = \mathcal{L}_{struct}$) in the numerator. The final implemented loss is:

$$\mathcal{L}_{uncer} = \sum_{\mathbf{p}} \left( \frac{\text{sg}[|r(\mathbf{p})|]}{2\boldsymbol{\beta}(\mathbf{p})^2} + \frac{1}{2}\ln \boldsymbol{\beta}(\mathbf{p}) \right). \qquad (16)$$

This modification ensures smoother gradient scaling, preventing numerical explosion, while preserving the outlier-rejection properties inherent to the uncertainty mechanism.

### B.2. Geometric Lifting for Flow-Induced Initialization

Here, we detail the geometric derivation for lifting 2D optical flow to 3D world velocity, ensuring notation consistency with Eq. 10 in the main text.

First, we recover the 3D point in the local camera coordinate system at frame $k$, denoted as $\mathbf{x}_{c,k}$, via pinhole back-projection using the estimated metric depth $d_i$ and intrinsic matrix $\mathbf{K}$:

$$\mathbf{x}_{c,k} = d_i \cdot \mathbf{K}^{-1}\tilde{\mathbf{u}}_i \qquad (17)$$

where $\tilde{\mathbf{u}}_i$ is the homogeneous pixel coordinate. To decouple object motion from camera ego-motion, we transform this point into the world frame using the camera pose $\mathbf{T}_k = [\mathbf{R}_k|\mathbf{t}_k]$:

$$\mathbf{x}_k = \mathbf{R}_k\mathbf{x}_{c,k} + \mathbf{t}_k. \qquad (18)$$

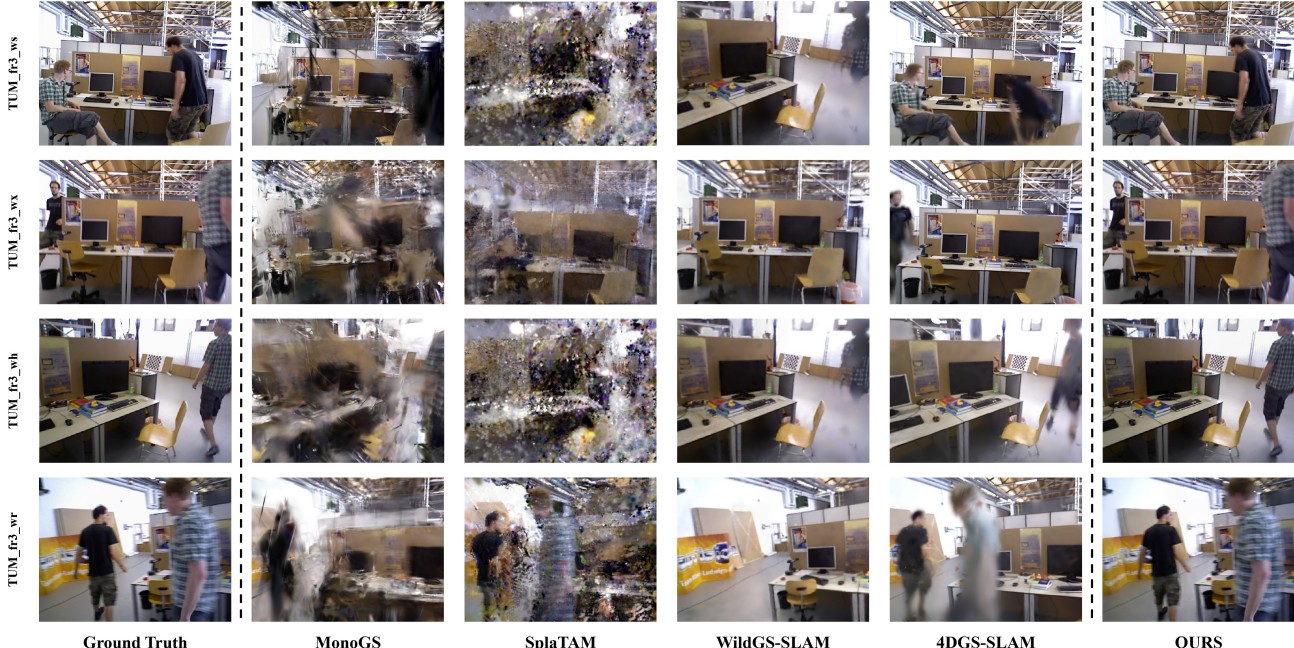

*Figure 5.* **Qualitative Comparison against Representative Baselines. MonoGS** suffers from severe ghosting in dynamic regions. **4DGS-SLAM** tends to over-smooth details. **De4D-SLAM** produces clearer reconstructions of both the static environment and dynamic agents.

*Table 14.* **Tracking Accuracy on the Full WildGS MoCap Split.** Per-sequence ATE RMSE [cm] ↓ on the ten-sequence WildGS MoCap benchmark (Zheng et al., 2025).

| Method | ANYmal1 | ANYmal2 | Ball | Crowd | Person | Racket | Stones | Table1 | Table2 | Umbrella | Avg. |
|--------|---------|---------|------|-------|--------|--------|--------|--------|--------|----------|------|
| WildGS-SLAM | **0.2** | 0.3 | 0.2 | **0.3** | 0.8 | **0.4** | **0.3** | **0.6** | **1.3** | **0.2** | **0.46** |
| De4D-SLAM | **0.2** | **0.2** | **0.1** | **0.3** | **0.7** | **0.4** | **0.3** | 0.7 | 1.6 | **0.2** | 0.47 |

Next, establishing temporal correspondence via the dense optical flow field $\mathbf{f}_{k \to k-1}$, we identify the corresponding pixel $\mathbf{u}_i'$ in the previous frame $k-1$. Crucially, we explicitly reject the constant depth assumption. Instead, we sample the metric depth map of the previous frame $D_{k-1}$ to obtain the conjugate depth $d_i' = D_{k-1}(\mathbf{u}_i')$. The previous world position $\mathbf{x}_{k-1}$ is derived similarly using $d_i'$ and $\mathbf{T}_{k-1}$.

Finally, the initialized 3D velocity vector $\mathbf{v}_{init} \in \mathbb{R}^3$ is computed as the finite difference over the interval $\Delta t$:

$$\mathbf{v}_{init} = \frac{\mathbf{x}_k - \mathbf{x}_{k-1}}{\Delta t}. \tag{19}$$

This explicitly anchors the dynamic Gaussian primitives with a physical velocity prior, preventing the non-convex optimization from stalling in zero-velocity local minima.

### B.3. Theoretical Justification for KAN over MLP

Our choice of the Kolmogorov-Arnold Network (KAN) over a traditional Multi-Layer Perceptron (MLP) is grounded in the approximation theory of high-frequency functions.

The motion boundaries in a dynamic scene represent high-frequency signals in the image domain (i.e., sharp transitions between static background and moving foreground).

**Spectral Bias.** Standard MLPs with ReLU activations suffer from *spectral bias*, tending to learn low-frequency components first and struggling to capture sharp discontinuities without excessive depth. In contrast, KANs employ learnable B-spline activation functions $\phi(x)$ on edges. According to the Kolmogorov-Arnold representation theorem, multivariable functions can be represented as superpositions of continuous univariate functions. The localized nature of B-splines allows KANs to adjust to local irregularities (motion boundaries) without affecting the global function approximation.

**Empirical Efficiency.** Under the specific architecture and training settings used in this work, the selected SA-KAN configuration uses 62.8% fewer parameters than the compared MLP baseline, while the quantitative ablation in Table 16 and the qualitative evidence in Fig. 8 support its

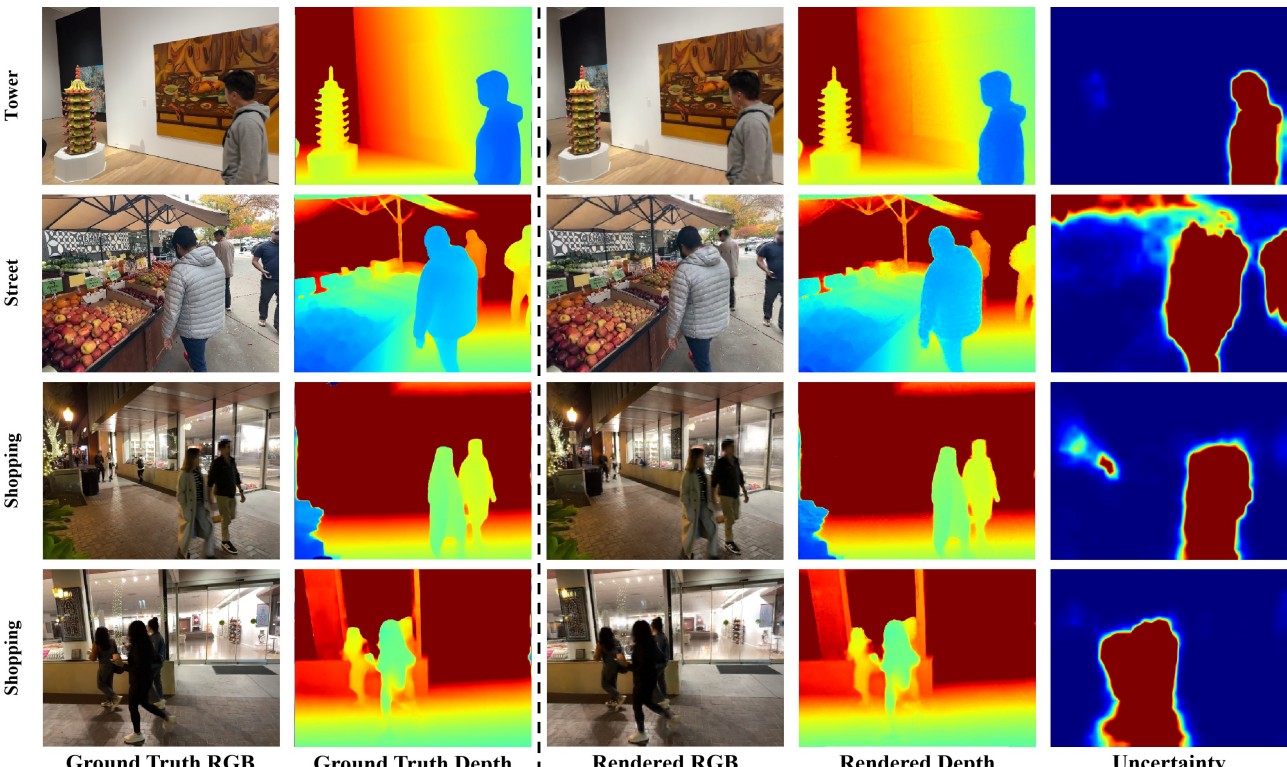

*Figure 6.* **Qualitative Generalization on the WildGS iPhone Split.** Representative results on mixed indoor and outdoor consumer captures without trajectory ground truth. From left to right, we show the ground-truth RGB image, ground-truth depth, rendered RGB, rendered depth, and the predicted uncertainty map. De4D-SLAM preserves coherent rendering and depth structure while assigning high uncertainty to moving foreground regions.

effectiveness for uncertainty-guided motion discovery in this task.

## C. Additional Benchmarking

In this section, we provide additional benchmark results to complement the comparisons in the main paper. We first report sequence-wise reconstruction metrics on the TUM RGB-D and Bonn RGB-D benchmarks. We then further evaluate generalization on the full WildGS MoCap split (Zheng et al., 2025), which provides a broader dynamic benchmark beyond the datasets used in the main paper. Finally, we include qualitative results on the WildGS iPhone split (Zheng et al., 2025), which contains mixed indoor and outdoor casual captures without trajectory ground truth.

### C.1. Reconstruction Benchmarks on TUM and Bonn

Tables 12 and 13 report the sequence-wise reconstruction metrics on the TUM RGB-D and Bonn RGB-D benchmarks. These results complement the average metrics reported in the main paper and provide a more detailed view of per-sequence behavior. Across both datasets, De4D-SLAM con-

sistently maintains strong structural fidelity and perceptual quality, particularly in sequences with substantial dynamic motion and occlusion. The quantitative results are further supported by Fig. 5, which illustrates that our method produces sharper holistic reconstructions and more coherent static-dynamic separation than the compared baselines.

### C.2. Generalization on the Full WildGS MoCap Split

To further assess generalization beyond TUM and Bonn, we evaluate De4D-SLAM on the full WildGS MoCap split (Zheng et al., 2025). This split contains ten motion-capture sequences and provides trajectory ground truth for quantitative tracking evaluation. As summarized in Table 14, De4D-SLAM maintains tracking robustness comparable to WildGS-SLAM (Zheng et al., 2025) on this broader benchmark. At the same time, our method preserves the ability to reconstruct dynamic scene content within a single framework, rather than removing dynamic regions as outliers. This observation is consistent with the main-paper comparisons: the proposed framework preserves localization performance while extending the output from static-only distractor removal to holistic dynamic reconstruction.

*Table 15.* **Data for Decoupling Analysis: Tracking Accuracy on Bonn** (ATE RMSE [cm] ↓).

| Variant | Balloon | Bal.2 | Person | Per.2 | Crowd | Crwd.2 | Sync | Sync2 | Rem. | Rem.2 | Place | Place2 | Avg. |
|---|---|---|---|---|---|---|---|---|---|---|---|---|---|
| w/o Decoupling | 5.38 | 2.39 | 4.13 | 6.94 | 2.53 | 4.06 | 0.74 | 0.97 | 1.62 | 2.06 | 1.74 | 1.90 | 2.87 |
| **Full Model** | **2.62** | **2.35** | **3.42** | **3.02** | **1.42** | **2.24** | **0.72** | **0.55** | **1.56** | **2.07** | **1.59** | **1.78** | **1.95** |

*Figure 7.* **Visual ablation of gradient-isolated decoupling on static-dynamic separation and reconstruction quality.** The red numbers overlaid on the rendered images represent the PSNR (dB). **w/o Decoupling:** Without the proposed strategy, the model suffers from significant ghosting artifacts in dynamic regions, and the uncertainty map fails to capture the dynamics (predominantly blue). **Ours:** Our method produces sharper renderings with higher PSNR values and correctly identifies dynamic objects in the uncertainty map (highlighted in red/yellow), demonstrating effective static-dynamic decoupling.

## C.3. Qualitative Generalization on the WildGS iPhone Split

The WildGS iPhone split (Zheng et al., 2025) does not provide trajectory ground truth and is therefore unsuitable for quantitative ATE evaluation. Nevertheless, it offers a broader set of casual dynamic captures spanning both indoor and outdoor scenes. As illustrated in Fig. 6, De4D-SLAM maintains coherent rendering, depth reconstruction, and uncertainty localization across these qualitatively diverse sequences.

## D. In-Depth Mechanism Analysis

In this section, we pair quantitative ablation data directly with visual evidence to provide a comprehensive analysis of the failure modes exhibited by baseline variants. We structure this analysis into three cohesive modules, detailing both the empirical results and the theoretical justifications for our architectural choices.

### D.1. Mechanism of Gradient-Isolated Decoupling

The **w/o Decoupling** variant simulates the naive joint optimization strategy commonly adopted in concurrent works.

**Theoretical Insight:** Without the gradient barrier, the optimization landscape becomes ill-conditioned. The highly flexible dynamic Gaussian primitives can rapidly absorb residual errors that should ideally be attributed to the static background. This creates a detrimental feedback loop: dynamic Gaussians expand to cover static regions to minimize photometric loss, which in turn compels the uncertainty network to classify these regions as dynamic (a phenomenon we term mask collapse).

**Empirical Consequence:** As shown in Table 15, this leads

*Table 16.* **Data for SA-KAN Analysis: Tracking Accuracy on Bonn** (ATE RMSE [cm] ↓).

| Variant | Balloon | Bal.2 | Person | Per.2 | Crowd | Crwd.2 | Sync | Sync2 | Rem. | Rem.2 | Place | Place2 | Avg. |
|---|---|---|---|---|---|---|---|---|---|---|---|---|---|
| w/o SA-KAN (MLP) | **2.51** | 2.51 | 3.96 | 3.06 | 1.55 | 2.16 | 0.78 | 0.58 | **1.26** | **1.93** | 1.82 | 1.98 | 2.01 |
| w/o SA (KAN) | 2.69 | **2.35** | 3.61 | 3.05 | 1.53 | **2.13** | 0.74 | **0.54** | 1.46 | 1.97 | 1.64 | 1.80 | 1.96 |
| **Full Model** | 2.62 | **2.35** | **3.42** | **3.02** | **1.42** | 2.24 | **0.72** | 0.55 | 1.56 | 2.07 | **1.59** | **1.78** | **1.95** |

*Figure 8.* **Visual Ablation of Uncertainty Module (SA-KAN vs. MLP).** The red numbers overlaid on the rendered images represent the PSNR (dB). **Left (w/o SA-KAN):** The pixel-wise MLP generates noisy, incoherent masks, erroneously suppressing valid static features on the background wall. **Right (Ours):** The SA-KAN produces smooth, spatially coherent masks that strictly adhere to the object boundaries.

to a clear degradation in tracking performance (Avg ATE 2.87 cm vs. 1.95 cm). Visually (Fig. 7), the system loses the ability to distinguish moving agents from the static structure, resulting in severe ghosting artifacts and map corruption.

### D.2. Impact of SA-KAN on Uncertainty Estimation

We rigorously verify the necessity of the Spatially-Aware KAN by comparing it against a standard pixel-wise MLP baseline.

**Theoretical Insight:** Standard MLPs process each pixel in isolation, lacking the geometric inductive bias required to interpret local context. While they can learn color-based segmentation, they struggle significantly with texture-less regions or complex lighting changes, resulting in high-frequency aleatoric uncertainty (noise). The SA-KAN addresses this limitation by integrating a spatial aggregation layer (DW-Conv) with Kolmogorov-Arnold spline activations. These learnable, non-linear splines adapt more effectively to the complex, non-convex boundaries of moving objects compared to fixed ReLU functions.

**Empirical Consequence:** As shown in Table 16, the **w/o SA-KAN** variant exhibits degraded tracking performance. The visual ablation (Fig. 8) confirms that the MLP produces

salt-and-pepper noise, erroneously suppressing valid static features (e.g., texture on walls), which dilutes the geometric constraints available for Bundle Adjustment.

### D.3. Geometric Convergence via Flow-Induced Initialization

The cold start problem represents a critical bottleneck in monocular 4D reconstruction.

**Theoretical Insight:** Optimizing the position and deformation of 4D Gaussians is a highly non-convex problem. Initializing dynamic primitives with zero velocity places the optimization starting point in a flat region of the energy landscape, making it difficult for gradient descent to discover the correct motion vector, particularly for fast-moving objects. By lifting 2D optical flow into 3D space, we provide a more favorable initialization for the optimizer, effectively guiding the Gaussian primitives along the physically correct trajectory from the very first iteration.

**Empirical Consequence:** Our quantitative analysis (Table 17 and Table 18) demonstrates a significant degradation in mapping fidelity for the **w/o Flow Init** variant (e.g., TUM Avg PSNR drops from 27.59 dB to 20.33 dB). Visually (Fig. 9), zero-velocity initialization manifests as amorphous,

*Table 17.* **Data for Initialization Analysis: Reconstruction Quality on TUM** (PSNR [dB] ↑, SSIM ↑, and LPIPS ↓).

| Variant | Metric | sitting_static | sitting_xyz | walking_static | walking_xyz | walking_rpy | walking_halfsphere | Avg. |
|---|---|---|---|---|---|---|---|---|
| w/o Flow Init | PSNR ↑ | 24.63 | 22.43 | 19.71 | 18.32 | 18.02 | 18.88 | 20.33 |
| | SSIM ↑ | 0.876 | 0.822 | 0.799 | 0.716 | 0.697 | 0.722 | 0.772 |
| | LPIPS ↓ | 0.147 | 0.208 | 0.225 | 0.309 | 0.358 | 0.324 | 0.262 |
| **Full Model** | PSNR ↑ | **28.64** | **25.14** | **27.84** | **27.11** | **28.78** | **28.03** | **27.59** |
| | SSIM ↑ | **0.924** | **0.877** | **0.915** | **0.912** | **0.917** | **0.919** | **0.911** |
| | LPIPS ↓ | **0.116** | **0.183** | **0.128** | **0.173** | **0.193** | **0.192** | **0.164** |

*Table 18.* **Data for Initialization Analysis: Reconstruction Quality on Bonn** (PSNR [dB] ↑, SSIM ↑, and LPIPS ↓).

| Variant | Metric | Balloon | Bal.2 | Person | Per.2 | Crowd | Crwd.2 | Sync | Sync2 | Rem. | Rem.2 | Place | Place2 | Avg. |
|---|---|---|---|---|---|---|---|---|---|---|---|---|---|---|
| w/o Flow Init | PSNR ↑ | 22.31 | 19.92 | 22.37 | 20.60 | 17.74 | 17.81 | 20.43 | 18.51 | 21.82 | 22.66 | 22.29 | 22.39 | 20.74 |
| | SSIM ↑ | 0.858 | 0.823 | 0.876 | 0.872 | 0.775 | 0.765 | 0.723 | 0.725 | 0.829 | **0.876** | 0.861 | 0.853 | 0.820 |
| | LPIPS ↓ | 0.265 | 0.289 | 0.246 | 0.240 | 0.336 | 0.352 | 0.370 | 0.364 | 0.298 | 0.245 | 0.249 | 0.283 | 0.295 |
| **Full Model** | PSNR ↑ | **27.31** | **28.43** | **27.49** | **27.15** | **27.49** | **28.78** | **27.88** | **23.28** | **28.66** | 22.79 | **25.24** | **25.89** | **26.70** |
| | SSIM ↑ | **0.929** | **0.934** | **0.929** | **0.933** | **0.921** | **0.930** | **0.919** | **0.822** | **0.934** | 0.795 | **0.879** | **0.914** | **0.903** |
| | LPIPS ↓ | **0.173** | **0.166** | **0.191** | **0.174** | **0.193** | **0.192** | **0.198** | **0.273** | **0.191** | **0.198** | **0.245** | **0.215** | **0.200** |

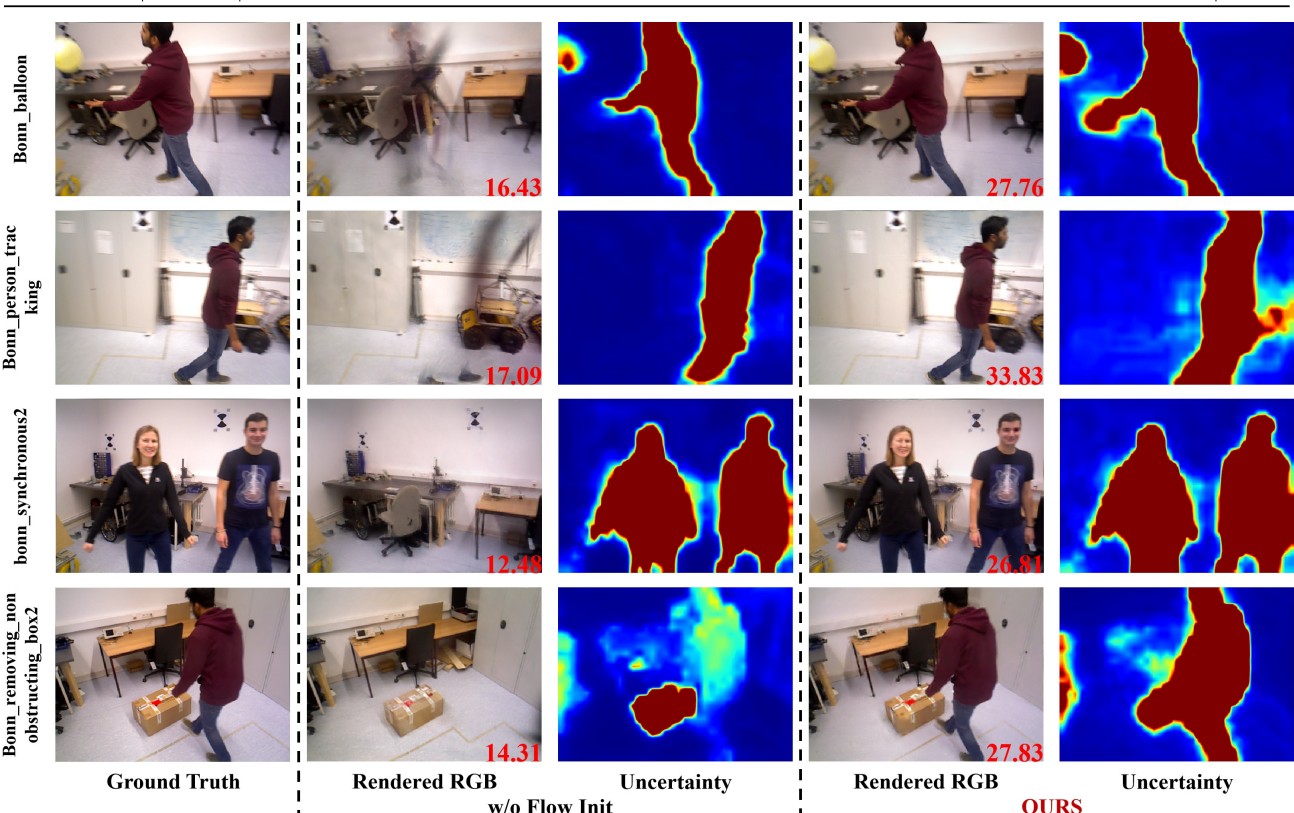

*Figure 9.* **Necessity of Flow-Induced Initialization.** The red numbers overlaid on the rendered images represent the PSNR (dB). **w/o Flow Init:** Trapped in local minima (blurred artifacts). **Ours:** Correct convergence to sharp geometry.

cloud-like artifacts where limbs are disconnected, whereas our method helps the model converge to sharper and more physically plausible geometry.

# E. Limitations and Future Work

**Limitations.** Despite the gains demonstrated in this work, our current system is not yet real-time and is best suited to high-end GPUs, as it jointly maintains tracking and dense dynamic Gaussian reconstruction within a unified pipeline. In addition, the method relies on the quality of upstream optical flow and on sufficient static structure to supervise uncertainty estimation; severe motion blur, heavy occlusion, or scenes dominated by moving objects may weaken tracking and downstream reconstruction. In particular, very fast motion or degraded flow estimation can reduce the reliability

of flow-induced initialization and make dynamic primitive activation less stable.

**Future Work.** Future research will prioritize efficiency and robustness. To further improve efficiency, we aim to optimize the dynamic Gaussian optimization process and reduce the overall computational burden without sacrificing reconstruction quality. To enhance robustness against visual degradations such as motion blur and heavy occlusion, we plan to develop stronger monocular motion priors and more stable uncertainty-guided optimization strategies. More broadly, we will continue to improve scalability for longer sequences and larger environments while preserving the benefits of joint tracking and holistic dynamic reconstruction.

