# OpenReview forum: "De4D-SLAM: Gradient-Isolated Static-Dynamic Decoupling for Monocular SLAM in Dynamic Environments"
_ICML.cc/2026/Conference — ICML 2026 regular_

### Official Review · Reviewer_FBmo · 2026-03-10

**Soundness:** 3
**Presentation:** 3
**Significance:** 3
**Originality:** 3
**Overall Recommendation:** 4
**Confidence:** 3

**Summary:**

This paper proposes De4D-SLAM, a monocular SLAM system for dynamic environments using a 4D Gaussian Splatting representation. Instead of treating dynamics as outliers, it reconstructs both static backgrounds and moving objects holistically. The method introduces a dual-stream static/dynamic map, a mathematically defined gradient-isolated optimization scheme (using stop-gradient operators), a Spatially-Aware KAN (SA-KAN) for uncertainty prediction, and flow-guided initialization. Experiments on the TUM and Bonn benchmarks demonstrate competitive tracking accuracy and high-quality rendering.

**Compliance With Llm Reviewing Policy:**

Affirmed.

**Final Justification:**

Most of my concerns have been satisfactorily addressed. I' ll maintain my positive score.

**Key Questions For Authors:**

Please refer to the weakness I mentioned.

**Limitations:**

Yes.

**Strengths And Weaknesses:**

Strengths:
- Meaningful Problem Framing: Tackling monocular SLAM in dynamic scenes without relying on category-specific semantic priors is a genuinely difficult and important problem. The shift from "masking dynamics away" to "joint reconstruction" is a strong conceptual motivation.
- Compelling Qualitative Results: The visual evidence, particularly in Figures 3 and 4, clearly illustrates the benefits of the proposed decoupling strategy. It successfully avoids the "holes" left by exclusion-based methods and the motion blur seen in naive joint optimization.
- Solid Empirical Performance: The method achieves strong results on standard dynamic SLAM benchmarks (e.g., best average ATE on Bonn), and the ablation studies validate the importance of key components like flow initialization.
- System Design: The overall architecture seamlessly integrates uncertainty-guided tracking with dual-stream mapping.

Weaknesses:
- Baseline Transparency and Fairness: The paper emphasizes "monocular" SLAM, yet the comparison protocol in Tables 1-3 does not explicitly state the input modalities of the baselines. Methods like SplaTAM and DynaSLAM are frequently run with RGB-D inputs. If De4D-SLAM (Monocular) is compared against RGB-D baselines without explicit labeling, the leaderboard is misleading.
-  Lack of Quantitative Evaluation for SA-KAN Masks: The uncertainty module (SA-KAN) is a central contribution. While the masks look qualitatively excellent in the figures, there is no quantitative evaluation of the motion segmentation accuracy. Tracking ATE alone does not fully validate the mask's precision.
- The analysis oversells some results: In Table 1, the average ATE improvement over WildGS is very small, and the method is not uniformly best on every sequence. In Table 3, the margin is again small. This is still good performance, but the surrounding text uses language closer to “clear dominance” than the numbers justify. The paper should present these tradeoffs more honestly.
-  System Overheads Placement: While the authors discuss the SA-KAN latency (2.57ms) and memory limits (<8GB VRAM) in the Appendix, a system paper of this complexity should explicitly report the end-to-end average tracking and mapping FPS.

---

> ### Author Rebuttal · Authors · 2026-03-31
>
> We sincerely thank the reviewer for the constructive feedback and for recognizing the motivation, qualitative results, and overall system design of our work. We address the concerns below.
>
> **Baseline transparency and modalities (W1):** We thank the reviewer for pointing out this ambiguity. You are correct that several baselines such as SplaTAM and DynaSLAM, were run in their default RGB-D settings, whereas De4D-SLAM operates in a strictly monocular setting using only RGB video as input. We agree that this should be made explicit in the paper. In the revision, we will clearly label the sensor modality of each method (e.g., `[RGB]` vs. `[RGB-D]`) in all comparison tables to make the evaluation protocol fully transparent.
>
> **Quantitative evaluation of SA-KAN masks (W2):** We agree that direct quantitative evaluation of the uncertainty/motion masks would be ideal. However, pixel-accurate dynamic masks are not natively available across the full tracking trajectories of the TUM and Bonn SLAM benchmarks. We therefore used downstream SLAM sensitivity as an indirect but practically meaningful validation. In our pipeline, these masks directly modulate which regions are down-weighted during frontend DBA. If the masks are inaccurate, especially with false positives on static background, the tracker loses reliable static constraints. Consistent with this, our ablation shows that replacing SA-KAN with a standard MLP degrades ATE from 1.27 cm to 1.38 cm, and Fig. 4 shows visibly noisier masks with stronger background bleeding. We will clarify in the revision that this is indirect evidence rather than a full segmentation benchmark.
>
> **Framing of results and comparison to WildGS (W3):** We agree with the reviewer that some of the original wording overstates the numerical margins. In particular, our tracking improvements over strong recent baselines such as WildGS are small, and our method is not uniformly best on every single sequence. We will revise the manuscript to use more accurate phrasing such as “highly competitive” rather than stronger claims of dominance. At the same time, we would like to emphasize the functional distinction: WildGS reconstructs only the static background by filtering dynamics, whereas De4D-SLAM achieves comparable tracking accuracy while jointly reconstructing both static and dynamic scene content. We will make this trade-off clearer in the revision.
>
> **End-to-end system overheads (W4):** We agree that end-to-end runtime should be reported more prominently in the main text. Our complete system runs at approximately **0.47 FPS** (**2.14 s/frame**), with the dominant bottlenecks coming from periodic local DBA and backend 4DGS map optimization rather than from SA-KAN itself. We already provide a more detailed component-wise breakdown in our response to **Reviewer ckL6**, and we will add this end-to-end runtime summary to the main evaluation section in the final version. The table below shows the results of our ablation study.
>
> |                    Module                    | Avg. online time |
> | :------------------------------------------: | :--------------: |
> |               SA-KAN inference               |  ~2.57 ms/call   |
> |          DINOv2 feature extraction           |  ~7.1 ms/frame   |
> |           Optical flow computation           |  ~13.3 ms/frame  |
> |         Local DBA pose optimization          | ~517.8 ms/frame  |
> | Backend dynamic-static Gaussian optimization | ~1224.6 ms/frame |
>
> We thank the reviewer again for the helpful suggestions.

---

> > ### Author Rebuttal · Reviewer_FBmo · 2026-04-04
> >
> > I would like to thank the authors for their detailed clarifications and explanations, which have addressed most of my concerns. At this stage, I have decided to maintain my current score and will further consider the other reviewers' feedback during the discussion phase.

---

> > > ### Author Response · Authors · 2026-04-08
> > >
> > > Thank you very much for your time, careful reading, and constructive feedback. We sincerely appreciate your thoughtful consideration throughout the discussion.
> > >
> > > Best regards,
> > >
> > > The Authors

---

### Official Review · Reviewer_24Wj · 2026-03-12

**Soundness:** 3
**Presentation:** 3
**Significance:** 2
**Originality:** 2
**Overall Recommendation:** 3
**Confidence:** 3

**Summary:**

This paper proposes De4D-SLAM, a monocular SLAM framework designed to perform joint static-dynamic reconstruction without categorical priors. The authors identify a problem, called the “optimization paradox”, which is when jointly optimizing static and dynamic Gaussian primitives without supervision, dynamic components tend to overfit static residuals, causing mask collapse and preventing motion discovery. To address this, the paper proposes: 1. Gradient-Isolated Static-Dynamic Decoupling A dual-stream architecture that prevents gradients from the static reconstruction stream from corrupting the dynamic stream. 2. SA-KAN: A lightweight network used to infer motion probability fields from visual features and activate dynamic primitives. 3. Flow-Induced Initialization: Optical-flow-based initialization to stabilize the optimization of 4D Gaussian primitives. Experiments on TUM and Bonn datasets show improvements in ATE tracking accuracy and rendering quality (PSNR / LPIPS) compared to existing dynamic SLAM baselines.

**Compliance With Llm Reviewing Policy:**

Affirmed.

**Key Questions For Authors:**

1. Have you compared SA-KAN against a spatially augmented MLP instead of a standard MLP?
2. As for linear Motion Model Limitations: how many short-lived Gaussian primitives are typically spawned per object for fast non-rigid motions (e.g., waving arms)? Is there a primitive count analysis showing the overhead of the piecewise linear approximation?
3. What is the average processing time per keyframe, including all components (DINOv2 feature extraction, SA-KAN, flow lifting, 4DGS optimization)?

**Limitations:**

yes

**Strengths And Weaknesses:**

**Well-Motivated Core Problem**. The Optimization Paradox (“Dynamic primitives overfit static residuals in unsupervised optimization”) is a genuine, well-articulated failure mode that has not been explicitly formalized in prior work. The theoretical characterization via the capacity mismatch between static and dynamic streams (Eq. 3–4) is clean and the intuition of "uncertainty collapse" is compelling. Figure 1 effectively illustrates the problem.

**Strong Quantitative Results**. On the Bonn dataset, the method achieves best average ATE and best average SSIM and LPIPS among methods that attempt holistic reconstruction. The claim of resolving the tracking-reconstruction tradeoff is well-supported.

**Thorough Ablations**. The paper provides unusually detailed ablations — per-sequence results on both TUM and Bonn for each component (Tables 9–12), paired with qualitative evidence (Figures 4, 6, 7, 8). The three-way decomposition (decoupling, SA-KAN architecture, flow initialization) clearly attributes gain to each component.

**Practical Design Choices**. The stop-gradient operator applied to the static residual in Eq. 7 is an elegant and minimal intervention to break the detrimental feedback loop. Flow-Induced Initialization is physically grounded and directly addresses a known non-convexity. The derivation in Appendix B.2 is complete. The piecewise linear trajectory model (Eq. 1–2) combined with temporal opacity windowing provides a reasonable approximation of non-rigid motion without requiring heavy deformation networks



# Weakness
**Novelty of the architecture is unclear**: several components appear incremental rather than fundamentally new. The paper should clarify how this differs from existing decoupling strategies in dynamic NeRF/SLAM.

**SA-KAN justification is weak**: Using Kolmogorov-Arnold Networks is fashionable, I didn’t see why KAN is better than MLP for this task. The theoretical argument in Appendix B.3 claims a KAN layer provides "parameter efficiency of O(N²G) compared to O(N²) for MLPs". This is a little bit misleading — the KAN actually has more parameters per layer than a comparable MLP (by a factor of G, the grid size), not fewer. The efficiency comes from the smaller network used (10.75k vs 28.86k) due to architectural choices, not from KAN intrinsically being more parameter efficient. The spectral bias argument is plausible but not experimentally validated beyond a tracking ATE improvement of 0.06 cm over MLP (Table 10: 2.01 vs 1.95 cm), a marginal gain. Given the added latency (2.57 ms vs 0.30 ms, a ~8.5x slowdown per Table 5), the practical case for KAN over a spatially augmented MLP is not convincingly made.

**Linear Velocity Model Contradicts Claims of Arbitrary Non-Rigid Motion**. The paper repeatedly claims the ability to reconstruct "arbitrary non-rigid motions", yet the core dynamic representation is a linear velocity model (Eq. 1). The argument that temporal opacity windowing (Eq. 2) enables piecewise-linear approximation of non-linear trajectories (Sec. 3.1) is under-analyzed. No formal error bound is provided for how well this piecewise-linear scheme approximates curved trajectories (e.g., articulated human limbs, rotating objects). The temporal extent is learned but its interaction with the piecewise approximation quality is neither controlled nor analyzed. Furthermore, no experiments test scenarios with genuinely complex non-rigid deformations (e.g., deformable objects, fluid-like motion).

**Limited Experimental Scope and Incomplete Baselines**. Both TUM and Bonn are small-scale indoor RGB-D datasets with relatively simple dynamic patterns (walking humans, balloons).  No evaluation is conducted on outdoor scenes, autonomous driving datasets (e.g., KITTI, Waymo), or scenarios with diverse object categories.

---

> ### Author Rebuttal · Authors · 2026-03-31
>
> We sincerely thank the reviewer for the constructive feedback and for appreciating our formulation of the “Optimization Paradox” and the stop-gradient design. We address the questions below.
>
> **SA-KAN vs. spatially augmented MLP (KQ1, W2):** We agree that the practical efficiency gain should not be interpreted as KAN intrinsically having fewer parameters per layer than an MLP, but rather as arising from the smaller effective architecture used in our implementation (10.75k parameters for SA-KAN vs. 28.86k for the MLP baseline). To directly address whether the gain is simply from adding spatial information to an MLP, we additionally compared SA-KAN against both a standard MLP and a spatially augmented MLP (SA-MLP) on the Bonn dataset:
>
> | **Architecture**  | **Avg. ATE (cm) ↓** |
> | :---------------: | :-----------------: |
> |        MLP        |        2.01         |
> |      SA-MLP       |        2.04         |
> | **SA-KAN (Ours)** |      **1.95**       |
>
> SA-KAN gives the best ATE among the tested uncertainty predictors. Together with the cleaner uncertainty boundaries shown in Fig. 4, this suggests that the gain is not recovered by spatial augmentation alone. Since these masks directly modulate which regions are trusted by the frontend DBA, cleaner boundaries help preserve reliable static constraints and are associated with more stable pose estimation.
>
> **Overhead of the piecewise linear approximation (KQ2, W3):** In our formulation, the linear assumption applies only to the translational motion of each Gaussian primitive center over its valid temporal window. More complex non-rigid deformations are approximated by (i) temporal handoff between short-lived dynamic primitives and (ii) continuous optimization of each primitive’s anisotropic covariance. On the highly dynamic Bonn *Crowd* sequence, we obtain:
>
> |                   Quantity                    |                  Value                   |
> | :-------------------------------------------: | :--------------------------------------: |
> | New short-lived dynamic primitives / keyframe |                  ~2,389                  |
> | Recycled / deactivated primitives / keyframe  |                  ~425.8                  |
> |      Active short-lived primitive buffer      |                  ~4,743                  |
> |         Peak dynamic primitive count          | 215,732 (14.27% of total scene capacity) |
>
> Since *Crowd* typically contains 4–6 moving individuals, this corresponds roughly to ~400–600 newly spawned dynamic primitives per object per keyframe. These statistics suggest that the piecewise linear design remains tractable in practice. We will add this analysis in the revision.
>
> **Component-wise processing time (KQ3):** Our complete online system runs at **~0.47 FPS** (**~2.14 s/frame**) on the highly dynamic Bonn *Crowd* sequence. The dominant bottlenecks are the local DBA (517.8ms) and backend dynamic-static Gaussian optimization (1224.6ms), while SA-KAN (2.57ms), DINOv2 (7.1ms), and optical flow (13.3ms) contribute only a small fraction of the runtime. For the detailed component-wise breakdown, please refer to our response to **Reviewer ckL6**.
>
> **Novelty of the decoupling strategy (W1):** We agree that prior dynamic NeRF/SLAM methods often include some form of static/dynamic separation, but this is typically achieved through semantic priors, motion masks, or explicitly pre-separated dynamic regions. Our novelty claim is therefore not merely separating static and dynamic content, but the **optimization-level decoupling** used to address a specific failure mode: under shared photometric optimization, the flexible dynamic representation can suppress the static residual signal needed for category-agnostic motion discovery, leading to uncertainty/mask collapse. Our contribution is to explicitly prevent this feedback using gradient-isolated supervision (Eq. 7) and detached weighting (Eq. 9), without relying on semantic priors to pre-separate dynamic objects.
>
> **Outdoor datasets and autonomous driving (W4):** We agree that broader evaluation, especially on outdoor scenes, would strengthen the paper. Our current experiments focus on TUM and Bonn because they directly stress the dynamic failure mode targeted in this work. For forward-facing autonomous-driving monocular setups such as KITTI/Waymo, dense 4D reconstruction is further complicated by degenerate motion geometry (limited effective parallax and lateral baseline), which introduces additional challenges beyond the present scope. To partially assess outdoor generalization during the rebuttal period, we additionally tested De4D-SLAM on outdoor scenes from the Wild-SLAM (iPhone) dataset and observed qualitatively stable static/dynamic separation. Consistent with the rebuttal link restrictions, we provide an anonymous figure page with representative outdoor frame : [**Outdoor Frames**](https://anonymous.4open.science/r/gjhjkure-65DF).
>
> We thank the reviewer again for the helpful suggestions.

---

> > ### Author Rebuttal · Reviewer_24Wj · 2026-04-07
> >
> > Thank author for the rebuttal. However, it doesn't substantially change the picture; the core weaknesses -- marginal SA-KAN gains, no formal analysis of the linear motion approximation, and a narrow evaluation scope -- still persist.

---

> > > ### Author Response · Authors · 2026-04-08
> > >
> > > Thank you for the continued feedback. We understand that your remaining concerns are mainly about (1) whether SA-KAN brings meaningful gains beyond a spatially augmented MLP, (2) whether the linear motion model is too limited without a formal analysis, and (3) whether the evaluation scope remains narrow. We briefly clarify these points below.
> > >
> > > **On the practical gain of SA-KAN.** We agree that the benefit should not be interpreted as KAN being intrinsically more parameter-efficient per layer than an MLP. Our point is narrower: the observed gain is not recovered simply by adding spatial augmentation to an MLP. To directly test this, we compared three uncertainty predictors under the same setting on Bonn: MLP (2.01 cm), SA-MLP (2.04 cm), and SA-KAN (1.95 cm) average ATE. Thus, SA-KAN still gives the best ATE among the tested predictors. Together with the cleaner uncertainty boundaries in Fig. 4, this suggests that the gain is not merely from adding spatial context, but from a more suitable uncertainty predictor for preserving reliable static constraints in DBA.
> > >
> > > **On the linear motion approximation.** We agree that we did not provide a formal approximation bound during the rebuttal period. What we intended to clarify is that the linear assumption applies only to the translational motion of each Gaussian primitive center over its valid temporal window. More complex non-rigid deformation is not attributed solely to this center trajectory; instead, it is approximated through temporal handoff between short-lived dynamic primitives together with continuous optimization of primitive parameters, especially anisotropic covariance and opacity. In this sense, the design should be interpreted as a local piecewise-linear approximation at the primitive-center level, rather than as a claim that all non-rigid deformation is captured by a single global linear motion model. To make this practical aspect more explicit, we added primitive-count statistics on the highly dynamic Bonn *Crowd* sequence, showing that the design remains tractable in practice.
> > >
> > > **On evaluation scope.** We agree that broader evaluation would strengthen the paper. During the rebuttal period, we therefore extended evaluation to the full WildGS MoCap split (10 sequences) rather than a selected subset. This split is the appropriate benchmark for ATE because it provides motion-capture ground truth; on this split, WildGS obtains 0.46 cm average ATE and our method 0.47 cm, i.e., essentially matched tracking accuracy under a richer reconstruction objective. We also clarified that the outdoor Wild-SLAM (iPhone) split cannot support ATE claims because it does not provide trajectory ground truth; accordingly, we use it only for qualitative outdoor generalization analysis.
> > >
> > > **Overall clarification of contribution.** More broadly, our intent is not to claim a large tracking-only margin over masking-based baselines such as WildGS. Rather, the contribution is that, in a stricter monocular self-supervised dynamic SLAM setting, robust localization can be maintained while also reconstructing dynamic scene content that exclusion-based methods explicitly discard. We therefore view the key result not as “dominance in ATE,” but as evidence that this richer static-and-dynamic reconstruction objective can be made trainable while remaining competitive in localization.
> > >
> > > Thank you again for the constructive discussion.

---

### Official Review · Reviewer_aFHw · 2026-03-12

**Soundness:** 3
**Presentation:** 2
**Significance:** 3
**Originality:** 3
**Overall Recommendation:** 4
**Confidence:** 4

**Summary:**

The paper presents De4D-SLAM, a framework designed for decoupled 4D reconstruction from monocular video. The motivation is that prior SLAM methods often filter dynamic objects to improve performance, but this comes at the cost of losing the ability to represent obstacles. This paper proposes a gradient-isolated decoupling strategy within a dual-stream architecture to separate the static background from dynamic entities. Then the authors use a hybrid architecture to infer motion probability fields from visual features, which could be used to identify dynamic objects. In this way, both the static and dynamic components of the scene are handled simultaneously. In addition, the authors introduce a flow-induced initialization prior that uses dense optical flow to solve the non-convexity of monocular 4D optimization.

**Compliance With Llm Reviewing Policy:**

Affirmed.

**Final Justification:**

I don’t have any further questions about the rebuttal. I keep my score as Weak Accept.

**Key Questions For Authors:**

1. How much of the improvement comes from the decoupling strategy and dual-stream architecture, rather than from Metric3D depth priors and DINOv2 features?

2. How robust is the method to fast-moving objects?

3. How sensitive is the method to the quality of optical flow?

**Limitations:**

Yes

**Strengths And Weaknesses:**

Strengths

1. The paper is motivated by an interesting aspect. Instead of filtering out dynamic objects, the authors attempt to handle both the static and dynamic components of the scene simultaneously.

2. The author propose a technically reasonable decoupling mechanism for the problem setting.

3. Experiments show strong quantitative results on tracking and reconstruction.

Weaknesses

1. Although the paper includes image comparisons with baselines, it may be difficult for reviewers to assess the qualitative performance of temporal consistency and motion quality without supplementary videos.

2. The ablation study covers decoupling, SA-KAN, and flow-based initialization, but does not isolate the contributions of the Metric3D depth prior or DINOv2 features.

---

> ### Author Rebuttal · Authors · 2026-03-31
>
> We sincerely thank the reviewer for the constructive feedback and for recognizing the motivation, decoupling mechanism, and strong quantitative results of our paper. We address the questions below.
>
> **Ablation on priors (KQ1, W2):** To better disentangle the gains from our architectural design and those from the employed priors, we conducted additional ablations on the Bonn dataset: removing Metric3D depth (`w/o Metric3D`), replacing DINOv2 features with shallow RGB convolutions (`w/o DINOv2`), removing the dual-stream decoupling (`w/o Decoupling`), and the full model.
>
> |    Method Variant     | ATE (cm) ↓ | PSNR (dB) ↑ |  SSIM ↑   |  LPIPS ↓  |
> | :-------------------: | :--------: | :---------: | :-------: | :-------: |
> |    w/o Decoupling     |    2.87    |    23.10    |   0.817   |   0.242   |
> |     w/o Metric3D      |    2.42    |    24.73    |   0.864   |   0.266   |
> | w/o DINOv2 (RGB Conv) |    2.21    |  **27.23**  | **0.912** |   0.214   |
> | **Ours (Full Model)** |  **1.95**  |    26.70    |   0.903   | **0.200** |
>
> These results suggest that the largest performance gap comes from the proposed decoupling mechanism: removing decoupling causes the clearest degradation across tracking and rendering. At the same time, the priors provide complementary benefits. Metric3D mainly improves scale stability and tracking robustness, while DINOv2 improves feature robustness, with some metric trade-offs depending on the sequence and evaluation criterion. We will add these ablations in the revision.
>
> **Robustness to fast-moving objects (KQ2):** Our framework is designed to alleviate the cold-start problem caused by rapid motion through the proposed Flow-Induced Initialization, which lifts 2D optical flow into 3D metric velocity estimates and provides a warm start for newly activated dynamic primitives. This is particularly helpful for fast non-rigid motion, where naive 3DGS optimization often struggles to converge. Empirically, on TUM’s highly dynamic *walking_halfsphere* sequence, De4D-SLAM maintains robust tracking (ATE: 1.40 cm) together with strong rendering quality (28.03 dB), indicating that the method remains effective under substantial foreground motion.
>
> That said, there remains an inherent limit in purely monocular visual SLAM: if motion becomes so fast that it produces severe local blur, or if dynamic objects occupy most of the field of view for an extended period, the available static constraints become insufficient and both reconstruction quality and camera tracking can degrade.
>
> **Sensitivity to optical flow (KQ3):** Our method is more sensitive to global flow degradation than to local flow noise. For local flow errors around fast-moving objects, the SA-KAN typically predicts high uncertainty in these unreliable regions, which suppresses their influence on both mapping and tracking; the frontend can therefore continue relying on stable static background regions. In contrast, when flow quality deteriorates globally, for example due to strong ego-motion blur or very large occlusions, the static constraints needed by monocular tracking are weakened everywhere, and failures can occur. We view this as an inherent limitation of purely monocular visual SLAM rather than a failure specific to our decoupling mechanism. Incorporating high-frequency inertial priors (e.g., IMU) is a future work to improve robustness in such cases.
>
> **Qualitative temporal consistency (W1):** We agree that temporal consistency and motion quality are best assessed over time. Consistent with the rebuttal link restrictions, we do not provide external videos here. Instead, we provide an anonymous figure page with representative frame comparisons sampled from the same dynamic clips, which illustrate reduced ghosting and flickering over time relative to the baselines: [**Contrast_Frames**](https://anonymous.4open.science/r/gjhjkure-65DF). We will also strengthen the qualitative discussion in the revision.
>
> We thank the reviewer again for the helpful suggestions.

---

> > ### Author Rebuttal · Reviewer_aFHw · 2026-04-04
> >
> > Thank the authors for their detailed explanation. The authors addressed the questions I raised. My main concerns are about the qualitative performance, robustness and sensitivity. Given the limited space of the rebuttal, I understand that the authors are unable to elaborate further. I will maintain my weak accept score.

---

> > > ### Author Response · Authors · 2026-04-08
> > >
> > > Thank you very much for your time, careful reading, and constructive feedback. We sincerely appreciate your thoughtful consideration throughout the discussion.
> > >
> > > Best regards,
> > >
> > > The Authors

---

### Official Review · Reviewer_ckL6 · 2026-03-13

**Soundness:** 2
**Presentation:** 3
**Significance:** 3
**Originality:** 2
**Overall Recommendation:** 3
**Confidence:** 4

**Summary:**

This paper proposes a monocular visual SLAM method that explicitly distinguishes static and dynamic components in a scene. Building upon DROID-SLAM, the proposed approach employs SA-KAN to separate static and dynamic regions and adopts a flow-induced initialization strategy to improve optimization.

**Compliance With Llm Reviewing Policy:**

Affirmed.

**Final Justification:**

The authors sincerely responded to my first- and second-round comments. Based on the second-round response, most of my concerns have been resolved, especially those regarding the WildGS iPhone sequences. I also understand that real-time operation may not yet be the most urgent requirement for 3DGS-based SLAM in highly dynamic scenes. My judgment remains quite borderline.

However, I still have a negative view of this work. Although the paper achieves a significant improvement in view-synthesis metrics over WildGS, this improvement appears to result largely from the fact that the ground-truth rendered images contain dynamic objects, whereas WildGS does not render them. Of course, handling dynamic objects is the main contribution of this work. Nevertheless, I expect that the rendering quality would be quite similar if dynamic objects were excluded. The pose-estimation results also seem to support this point.

In my opinion, the only advantage of this work over WildGS is that it can model static and dynamic objects separately using different 3DGS maps. I am still uncertain about the practical usefulness of maintaining a separate 3DGS map for dynamic objects. Therefore, I would like to maintain my original decision of **weak reject** for this work.

**Key Questions For Authors:**

1. It would be helpful to provide the computational time, including the processing time for each submodule. This would allow readers to identify which components are the main bottlenecks for achieving real-time performance.

2. The threshold $\tau$ used to determine static and dynamic regions appears to be an important parameter. It would be helpful to analyze the sensitivity of this parameter over a range of values, including very large values (as an extreme case of assigning all pixels as static, which corresponds to the common assumption used in many traditional SLAM methods).

**Limitations:**

1. As mentioned in the weaknesses, the proposed method does not operate in real time, which is an important requirement for practical SLAM systems. Since the authors already acknowledge this issue and mention it in the future work section, it is expected to be addressed in future work.

**Strengths And Weaknesses:**

[Strengths]
1. The proposed method explicitly distinguishes static and dynamic components in the scene. The paper clearly identifies the optimization paradox that arises when jointly solving the SLAM and segmentation problems, and proposes a strategy to mitigate this issue (referred to as uncertain collapse).

[Weaknesses]
1. Real-time operation is a critical requirement for many SLAM systems. However, the proposed method does not operate in real time (this limitation is already mentioned by the authors in the future work section).

2. The experiments use only two datasets, although many SLAM datasets are available across both indoor and outdoor environments.

3. Figure 2 contains several ambiguous or unclear elements. First, there is no explicit component showing how static and dynamic pixels are distinguished. Second, the DBA module may include mapping functionality, but it is placed completely outside the mapping module in the diagram. Third, the input includes the next frame, which may lead readers to misunderstand that the proposed method uses future frames. If the next frame is not actually used in the current frame, it would be better to remove it from the figure.

---

> ### Author Rebuttal · Authors · 2026-03-31
>
> We sincerely thank the reviewer for recognizing our identification of the optimization paradox in joint static-dynamic learning. We also appreciate the constructive concerns regarding real-time performance, dataset breadth, Figure 2 clarity, and the sensitivity of the threshold  $\tau$.
>
> **Latency breakdown and real-time performance (W1, Lim1, KQ1):** We agree that real-time operation is important for many practical SLAM systems, and our current implementation does not yet meet that requirement. On the highly dynamic Bonn dataset, the complete online system runs at **~0.47 FPS** (**~2.14 s/frame**). To directly answer KQ1, we profiled the **major** online components and report their average runtime contribution below (normalized per input frame where applicable):
>
> |                    Module                    | Avg. online time |
> | :------------------------------------------: | :--------------: |
> |               SA-KAN inference               |  ~2.57 ms/call   |
> |          DINOv2 feature extraction           |  ~7.1 ms/frame   |
> |           Optical flow computation           |  ~13.3 ms/frame  |
> |         Local DBA pose optimization          | ~517.8 ms/frame  |
> | Backend dynamic-static Gaussian optimization | ~1224.6 ms/frame |
>
> This shows that the dominant bottlenecks are local DBA and backend 4DGS map optimization, rather than the modules introduced by our method. In particular, our learned components are lightweight relative to the full monocular 4DGS pipeline. We agree this is an important limitation. Our ongoing efforts to improve runtime focus on the dominant optimization bottlenecks identified above, including: (i) replacing the heavy local DBA with a lighter patch-based flow frontend, (ii) adopting a fully decoupled non-blocking sliding-window backend, and (iii) explicitly constraining the dynamic stream complexity (e.g., limiting SH degree / primitive growth) to bound optimization cost.
>
> **Sensitivity of $\tau$ (KQ2):** We agree that  $\tau$ , which controls dynamic primitive activation, is an important parameter. We evaluated four settings on the highly dynamic Bonn *Crowd* sequence:
>
> |     $\tau$      |  0.3  |  0.5  | 0.8 (default) |  1.0  |
> | :-------------: | :---: | :---: | :-----------: | :---: |
> | **ATE (cm) ↓**  | 1.51  | 1.52  |   **1.42**    | 1.49  |
> | **PSNR (dB) ↑** | 26.68 | 27.31 |   **27.49**   | 27.10 |
>
> Performance remains stable across a practical range of $\tau$ values, with $\tau=0.8$ giving the best overall trade-off. This indicates that the method is not overly brittle to threshold tuning. For the reviewer’s suggested extreme case, when  $\tau$  becomes very large, dynamic primitive activation is effectively suppressed, and the system approaches a **static-only variant**. As expected, performance drops substantially in dynamic scenes. We will add this sensitivity analysis in the revision.
>
> **Figure 2 clarification (W3):** Thank you for pointing out the ambiguity. Please refer to our updated anonymous Figure 2 here: [**System Overview**](https://anonymous.4open.science/r/gjhjkure-65DF). In the revision, we will: (i) make SA-KAN / the Learnable Disentanglement Module explicit as the source of the uncertainty map for static-dynamic separation; (ii) revise the DBA-mapping interaction so that feedback from backend uncertainty to frontend pose optimization is visually clear; and (iii) remove the potentially misleading “Next Frame” phrasing and revise the input illustration to emphasize that the pipeline is causal and processes frames sequentially, without using future observations for the current estimate.
>
> **Dataset diversity (W2):** We agree that broader evaluation is valuable. In the submission, we focused on the dynamic sequences of **TUM RGB-D** and **Bonn** because they directly stress the non-rigid motion and occlusion patterns relevant to our decoupling mechanism. Many standard SLAM benchmarks are predominantly static and thus do not stress-test the specific dynamic failure mode studied here as directly. To further assess generalization during the rebuttal period, **we additionally evaluated our method on the recent WildGS benchmark, using all sequences in the Mocap split and reporting average results**:
>
> | Method | ATE (cm) ↓ | PSNR (dB) ↑ |  SSIM  ↑  | LPIPS  ↓  |
> | :----: | :--------: | :---------: | :-------: | :-------: |
> | WildGS |  **0.46**  |    20.59    |   0.783   |   0.209   |
> |  Ours  |    0.47    |  **31.05**  | **0.946** | **0.183** |
>
> These results suggest that our method maintains comparable tracking robustness while achieving substantially better holistic rendering quality on a broader unseen dynamic benchmark. This is consistent with the design difference that WildGS masks dynamic entities to preserve a static background map, whereas our method explicitly reconstructs both the background and moving objects.
>
> We thank the reviewer again for the helpful suggestions.

---

> > ### Author Rebuttal · Reviewer_ckL6 · 2026-04-01
> >
> > I appreciate the authors' efforts to address my concerns. However, I remain skeptical regarding the overall contribution and value of this work for the following reasons:
> >
> > **1. Computational Overhead vs. Performance Gain:** The proposed method requires additional computation and time to model dynamic and static components separately. However, this segmentation does not seem to yield a significant "synergy" that improves tracking accuracy, which remains comparable to or only marginally better than WildGS-SLAM. Since the static/dynamic separation appears to serve only as a modeling choice without enhancing core SLAM metrics, the practical utility of providing rendered images with dynamic objects is questionable.
> >
> > **2. Evaluation on Diverse Datasets:** I expect the authors to include the full WildGS dataset, encompassing both MoCap and iPhone sequences. The iPhone sequences, in particular, appear to be more challenging due to their outdoor settings and complex backgrounds.

---

> > > ### Author Response · Authors · 2026-04-02
> > >
> > > Thank you for the continued feedback. We agree that the key question is whether the added static/dynamic modeling complexity brings meaningful value if the tracking improvement over a strong baseline such as WildGS is only modest.
> > >
> > > **On contribution framing,** we understand the reviewer’s current interpretation, and this may also be due to our not having expressed this point clearly enough in the previous version. Our intention is not to present the method simply as adding a dynamic rendering branch to a conventional tracker, but rather as addressing a stricter formulation of monocular dynamic SLAM, namely preserving tracking robustness while also reconstructing dynamic scene content that masking-based methods explicitly discard. We introduce gradient-isolated decoupling to make this joint objective trainable under self-supervision by addressing the uncertainty-collapse / optimization-paradox failure mode identified in the paper.
> > >
> > > **Regarding comparison with WildGS,** we agree with the reviewer that the tracking improvement is modest. At the same time, our method addresses a richer problem of reconstructing static and dynamic scenes rather than WildGS’s static-only distractor-removal objective. Under this harder setting, our tracking remains in the same performance regime as WildGS, with modest gains on Bonn/TUM and essentially matched accuracy on the full WildGS MoCap split. We therefore do not present this as a claim of large gains over WildGS, but as evidence that the richer reconstruction objective does not materially compromise localization robustness. Consistent with this point, our draft also includes comparisons with dynamic 4DGS systems, which further suggests that the proposed monocular self-supervised formulation is not only functionally meaningful, but also compares favorably with similar methods in terms of localization reliability and robustness.
> > >
> > > **On practical utility,** we also appreciate the concern that reconstructing dynamics should provide value beyond visualization. In dynamic environments, moving objects are part of the scene state rather than mere outliers. Masking-based methods explicitly remove this information, whereas our method retains both static background and moving agents, yielding a more complete scene representation for downstream perception, interaction, and scene reasoning.
> > >
> > > **Regarding runtime,** we agree that this is an important practical consideration. Our current implementation is not real-time. Since this work extends beyond pose estimation to joint dynamic mapping and reconstruction, we believe it is more appropriate to interpret runtime within the same class of Gaussian-Splatting-based SLAM methods. In this setting, profiling shows that the dominant cost comes from the shared dense optimization backbone, while SA-KAN and DINOv2 contribute only a small fraction of total runtime. We therefore believe this runtime trade-off is broadly comparable within similar Gaussian-Splatting-based SLAM methods while supporting a richer output.
> > >
> > > **On evaluation protocol and dataset diversity,** we appreciate the reviewer’s concern regarding dataset diversity and evaluation coverage, and we agree that broader evaluation would strengthen the paper. We therefore further clarify that our WildGS rebuttal result was evaluated on the full WildGS MoCap split (10 sequences), not a selected subset. This split has motion-capture ground truth and is therefore the appropriate benchmark for ATE evaluation; the WildGS iPhone split has no GT trajectories and is used only qualitatively. On the full MoCap split, the per-sequence ATEs are:
> > >
> > > | Method | ANYmal1 | ANYmal2 | Ball | Crowd | Person | Racket | Stones | Table1 | Table2 | Umbrella | Avg. |
> > > | ------ | ------- | ------- | ---- | ----- | ------ | ------ | ------ | ------ | ------ | -------- | ---- |
> > > | WildGS | 0.2     | 0.3     | 0.2  | 0.3   | 0.8    | 0.4    | 0.3    | 0.6    | 1.3    | 0.2      | 0.46 |
> > > | Ours   | 0.2     | 0.2     | 0.1  | 0.3   | 0.7    | 0.4    | 0.3    | 0.7    | 1.6    | 0.2      | 0.47 |
> > >
> > > Finally, we additionally tested De4D-SLAM on outdoor scenes from the Wild-SLAM (iPhone) dataset. Since this split has no GT trajectories, and WildGS itself uses it only qualitatively, we do not use it for ATE claims. Nevertheless, we observed qualitatively stable static/dynamic separation and outdoor generalization; anonymous visual results: [Outdoor Frames](https://anonymous.4open.science/r/gjhjkure-65DF)
> > >
> > > Overall, we understand that the paper may currently read as if it should be judged primarily by real-time throughput or by a large ATE margin over a masking-based baseline. What we hope to clarify, however, is that, in a self-supervised monocular framework, robust localization and holistic dynamic reconstruction can be achieved together, yielding a richer scene representation than static-only removal. We appreciate the reviewer’s questions, which helped us clarify this positioning.
> > >
> > > Thank you again for the constructive feedback.

---

### Decision · Program_Chairs · 2026-04-30

**Decision:**

Accept (regular)

**Comment:**

The paper received mixed reviews, with reviewers raising concerns about limited evaluations, the lack of ablations comparing SA-KAN layers to standard MLPs, and the absence of real-time performance analysis. In their rebuttal, the authors addressed these issues to a certain extent by providing additional clarifications and experiments. The work tackles the challenging problem of jointly reconstructing static and dynamic components of a video. Taking into account the manuscript, the reviews, and the rebuttal, the meta-reviewers conclude that the paper represents a valuable contribution to the community. The authors are requested to revise and improve the paper as per the discussions.